# Multimodal Bandits: Regret Lower Bounds and Optimal Algorithms

**William Réveillard**
Division of Decision and Control Systems
KTH Royal Institute of Technology
11428 Stockholm, Sweden
wilrev@kth.se

**Richard Combes**
Laboratoire des signaux et systèmes
Université Paris-Saclay, CNRS, CentraleSupélec
91190 Gif-sur-Yvette, France
richard.combes@centralesupelec.fr

## Abstract

We consider a stochastic multi-armed bandit problem with i.i.d. rewards where the expected reward function is multimodal with at most $m$ modes. We propose the first known computationally tractable algorithm for computing the solution to the Graves-Lai optimization problem, which in turn enables the implementation of asymptotically optimal algorithms for this bandit problem.

## 1 Introduction

We consider a stochastic multi-armed bandit with $K \geq 1$ arms. At time $t \in [T]$, with $T \geq 1$, based on her previous observations, a learner selects an arm $k(t)$ in $[K] = \{1, \ldots, K\}$, and subsequently observes a random reward $X_{k(t),t}$. The successive rewards $(X_{k,t})_{t \in \mathbb{N}}$ obtained when sampling a given arm $k \in [K]$ are drawn i.i.d. from a family of distributions $\nu_k(\mu_k)$ parameterized by their expectation $\mu_k$. The vector of mean rewards[1] $\boldsymbol{\mu} = (\mu_k)_{k \in [K]}$ is unknown to the learner. The learner's goal is to select arms in order to discover the optimal arm $k^\star(\boldsymbol{\mu}) = \arg\max_{k \in [K]} \mu_k$. More precisely, the learner aims at minimizing the regret

$$R(\boldsymbol{\mu}, T) = T(\max_{k \in [K]} \mu_k) - \sum_{t \in [T]} \mathbb{E}[\mu_{k(t)}]$$

which is the expected difference between the total reward obtained by an oracle who knows $\boldsymbol{\mu}$ in advance and always chooses the arm with the largest mean reward, and the total reward obtained by the learner. If we were to assume that $\boldsymbol{\mu}$ is arbitrary, then the problem at hand would reduce to the classical stochastic multi-armed bandit. Here we consider a *structured* stochastic multi-armed bandit, where the structure is encoded by a graph $G$.

We consider a graph $G = (V, E)$ whose vertices are the arms $V = [K]$, and we say that arms $k$ and $\ell$ are neighbors if and only if $(k, \ell) \in E$. Arm $k$ is a mode of $\boldsymbol{\mu}$ with respect to $G$ if and only if it has a strictly greater reward than all of its neighbors: $\mu_k > \max_{\ell:(k,\ell) \in E} \mu_\ell$ and we say that $\boldsymbol{\mu}$ is $m$-modal with respect to $G$ if it has $m$ modes with respect to $G$.

In this work, we assume that $\boldsymbol{\mu}$ is at most $m$-modal with respect to a graph $G$ known to the learner. We emphasize that, while $G$ is known to the learner, $\boldsymbol{\mu}$ is not, so that finding the optimal arm requires sampling from suboptimal arms repeatedly. We note that for $m = 1$, a 1-modal vector is simply a unimodal vector, thus the problem reduces to unimodal bandits [8]. If $G$ is a line graph, a mode is an arm whose reward is greater than that of its left and right neighbors. We will assume that $G$ is a tree.

---

[1] With a slight abuse of notation, we also refer to $\boldsymbol{\mu}$ as the reward function of the bandit problem.

39th Conference on Neural Information Processing Systems (NeurIPS 2025).

## 2  Related work and contribution

In the absence of a multimodal structure, our problem reduces to the classical multi-armed bandit studied by [13], and several asymptotically optimal algorithms are known for this problem, such as KL-UCB, proposed by [6], DMED, proposed by [9] and Thompson Sampling, as analyzed by [10].

When adding a multimodal structure with $m = 1$ mode, we obtain the unimodal bandit problem, originally studied by [8] and revisited by [25]. Several asymptotically optimal algorithms have been proposed for this problem including the KL-UCB style algorithm of [2], the Thompson Sampling style algorithm of [21] and the DMED style algorithm of [19]. A common feature of these algorithms is that they are all based on *local search*, where only arms that are neighbors of the optimal arm are selected a logarithmic amount of time. Local search is necessary for asymptotic optimality in unimodal bandits because the strategy that minimizes the Graves-Lai bound, as introduced by [7], is local.

Multimodal bandits with $m \geq 1$ modes generalize unimodal bandits, and have been considered in [17] and [18]. These works explored local search strategies, which, as we shall see, are not necessarily asymptotically optimal. The main motivation behind multimodal bandits is the fact that many objective functions encountered in applications are not convex nor unimodal, such as the empirical risk of deep neural networks. Methods for optimizing or sampling (which are closely related) multimodal functions include bayesian methods studied by [3] and MCMC methods considered by [14]. Multimodal functions have also been considered in an *active learning* setting by [16]. An interesting application of bandit problems with multimodal rewards is pricing ([15, 24]).

For structured bandits, the Graves-Lai bound is an information-theoretic regret lower bound, stated as an optimization problem. Its optimal solution identifies strategies for *optimal exploration*, i.e., the rate at which suboptimal arms must be selected to ensure minimal regret. Several asymptotically optimal algorithms with regret matching the Graves-Lai bound have been proposed by [7], [1], [5], [22]. The main advantage of these algorithms is their *universality* in the sense that they apply to all structured bandits.

While the above algorithms are indeed universally asymptotically optimal, they often pose a tremendous computational challenge, because they must solve the Graves-Lai optimization problem. In some simple structures, the Graves-Lai optimization problem admits a closed-form solution, for instance: classical bandits ([13]), unimodal bandits ([2]), dueling bandits ([11]) to name a few. For some other structures such as combinatorial bandits ([4]), the Graves-Lai optimization problem can be solved with efficient iterative algorithms. For multimodal bandits, solving the Graves-Lai optimization problem is challenging, as we shall see, primarily due to the highly non-convex nature of its constraint set.

**Our contribution.** In this work, we provide the first known computationally tractable algorithm to solve the Graves-Lai optimization problem for multimodal bandits. The algorithm is involved and uses a combination of discretization, dynamic programming and projected subgradient descent in order to navigate the intricate structure of the constraint set. The algorithm applies to a wide variety of reward distributions, and any tree graph. We further demonstrate that local search strategies are suboptimal, which means that solving the Graves-Lai problem is unavoidable for optimality. The code for the proposed algorithms, which are involved, is publicly available at `https://github.com/wilrev/MultimodalBandits`.

## 3  Asymptotically optimal algorithms for multimodal bandits

In this section, we state our problem assumptions, present the Graves-Lai lower bound specialized to the case of multimodal bandits, and recall how solving the Graves-Lai optimization problem enables one to design asymptotically optimal algorithms, i.e., with regret matching the Graves-Lai lower bound.

**Notation.**  To ease exposition, we use the following notation. All vectors are represented with bold symbols. We denote by $e^{(k)} \in \mathbb{R}^K$ the $k$-th canonical basis vector of $\mathbb{R}^K$, and by $\|x\|_p = \left(\sum_{k \in [K]} |x|^p\right)^{1/p}$ the $L_p$ norm. The closure of a set $S \subset \mathbb{R}^K$ is denoted by $\overline{S}$. We denote $\mu^\star = \max_{k \in [K]} \mu_k$ and $\mu_\star = \min_{k \in [K]} \mu_k$ the maximum and minimum mean reward, respectively. We

assume that the optimal arm $k^\star(\boldsymbol{\mu}) = \arg\max_{k \in [K]} \mu_k$ is unique. We define the vector of gaps $\boldsymbol{\Delta} = (\mu^\star - \mu_k)_{k \in [K]}$ and the minimal gap $\Delta_{\min} = \min_{k \in [K]: \Delta_k > 0} \Delta_k$.

For a given tree $G = (V, E)$ with $V = [K]$, we denote by $\operatorname{diam}(G)$ its diameter and $\deg(G)$ its maximal degree. $\mathcal{M}(\boldsymbol{\mu})$ is the set of modes of $\boldsymbol{\mu}$ with respect to $G$, so that $\boldsymbol{\mu}$ is $m$-modal if $|\mathcal{M}(\boldsymbol{\mu})| = m$, and $\mathcal{N}(\boldsymbol{\mu})$ is the set of modes and neighbors of modes of $\boldsymbol{\mu}$. We define $\mathcal{F}_{\leq m}$ (resp. $\mathcal{F}_m$) the set of reward functions of $\mathbb{R}^K$ with at most $m$ modes (resp. exactly $m$ modes) with respect to $G$. We assume that $\boldsymbol{\mu} \in \mathcal{F}_{\leq m}$ for some $m > 1$, and that $m$ is *known* to the learner.

Finally, we sometimes consider the tree $G$ to be directed. Then, for a given arm $k \in [K]$, we denote by $\mathcal{C}(k)$ the set of children of $k$, $\mathcal{D}(k)$ the set of descendants of $k$, $p(k)$ the parent of $k$ and $p^2(k)$ the grandparent of $k$ (i.e., the parent of $p(k)$).

**Assumptions on reward distributions.** The rewards from arm $k \in [K]$ form an i.i.d. sample from distribution $\nu_k(\mu_k)$. Let $d_k(\mu_k, \lambda_k) = D(\nu_k(\mu_k) \parallel \nu_k(\lambda_k))$ denote the relative entropy between the rewards distributions of arm $k$ under parameters $\mu_k$ and $\lambda_k$. For $\boldsymbol{\mu}, \boldsymbol{\lambda} \in \mathbb{R}^K$ we use the vectorized notation $d(\boldsymbol{\mu}, \boldsymbol{\lambda}) = (d_k(\mu_k, \lambda_k))_{k \in [K]}$. We make two assumptions regarding these relative entropies.

**Assumption 1.** *For all $\boldsymbol{\mu} \in \mathbb{R}^K$ and $k \in [K]$, $\lambda_k \mapsto d_k(\mu_k, \lambda_k)$ is strictly decreasing for $\lambda_k < \mu_k$ and strictly increasing for $\lambda_k > \mu_k$.*

**Assumption 2.** *For all $k \in [K]$, $\boldsymbol{\mu} \in \mathbb{R}^K$ and $\boldsymbol{\lambda}, \boldsymbol{\lambda}'$ in $[\mu_\star, \mu^\star]^K$ we have $\|d(\boldsymbol{\mu}, \boldsymbol{\lambda}) - d(\boldsymbol{\mu}, \boldsymbol{\lambda}')\|_1 \leq \mathfrak{A}(\boldsymbol{\mu})\|\boldsymbol{\lambda} - \boldsymbol{\lambda}'\|_1$ where $\mathfrak{A}(\boldsymbol{\mu}) \geq 0$ can be understood as the Lipschitz constant of the relative entropy when its first argument is held constant.*

The first assumption boils down to the relative entropy $d_k$ being unimodal, and its unique mode being the minimizer $\lambda_k = \mu_k$. The second assumption is satisfied whenever the divergence is continuously differentiable over $[\mu_\star, \mu^\star]^K$. For example, when $\nu_k(\mu_k) = \mathcal{N}(0, 1)$, it holds with $\mathfrak{A}(\boldsymbol{\mu}) = \mu^\star - \mu_\star$.

**Regret lower bound.** Proposition 1 states that the asymptotic regret of any uniformly good algorithm (i.e., whose regret scales subpolynomially on all problem instances) must be lower bounded by the value of the Graves-Lai optimization problem multiplied by $\ln T$. We denote this optimization problem by $P_{GL}$.

**Proposition 1.** *Consider an algorithm such that $\lim_{T \to \infty} \frac{R(\boldsymbol{\mu}, T)}{T^\alpha} = 0$ for all $\alpha > 0$ and all $\boldsymbol{\mu} \in \mathcal{F}_{\leq m}$. Then its asymptotic regret is lower bounded as $\liminf_{T \to \infty} \frac{R(\boldsymbol{\mu}, T)}{\ln T} \geq C(m, \boldsymbol{\mu})$ for all $\boldsymbol{\mu} \in \mathcal{F}_{\leq m}$ where $C(m, \boldsymbol{\mu})$ is the value of:*

$$\text{minimize}_{\boldsymbol{\eta}} \; \boldsymbol{\eta}^\top \boldsymbol{\Delta} \text{ subject to } \inf_{\boldsymbol{\lambda} \in \mathcal{B}(m, \boldsymbol{\mu})} \boldsymbol{\eta}^\top d(\boldsymbol{\mu}, \boldsymbol{\lambda}) \geq 1 \text{ and } \boldsymbol{\eta} \geq 0 \qquad (P_{GL})$$

$$\text{with } \mathcal{B}(m, \boldsymbol{\mu}) = \{\boldsymbol{\lambda} \in \mathcal{F}_{\leq m}, \lambda_{k^\star(\boldsymbol{\mu})} = \mu^\star, k^\star(\boldsymbol{\lambda}) \neq k^\star(\boldsymbol{\mu})\}.$$

$P_{GL}$ is a semi-infinite linear program. There are infinitely many constraints, and these constraints are described by $\mathcal{B}(m, \boldsymbol{\mu})$, which is the set of *confusing parameters*. $\boldsymbol{\lambda} \in \mathcal{B}(m, \boldsymbol{\mu})$ is *confusing* to the learner because it cannot be distinguished from $\boldsymbol{\mu}$ by selecting the optimal arm $k^\star(\boldsymbol{\mu})$, thereby forcing the learner to explore suboptimal arms. In particular, for a fixed $\boldsymbol{\eta} \geq 0$, we call *most confusing parameter* the minimizer $\boldsymbol{\lambda}^\star$ of $\boldsymbol{\lambda} \mapsto \boldsymbol{\eta}^\top d(\boldsymbol{\mu}, \boldsymbol{\lambda})$ over $\overline{\mathcal{B}}(m, \boldsymbol{\mu})$. The set $\mathcal{B}(m, \boldsymbol{\mu})$ has a complicated structure, which is why solving $P_{GL}$ is non-trivial. Proposition 1 is a direct consequence of the general bound of Theorem 1 in [7] or alternatively the simpler lower bound of Theorem 1 in [1].

**Asymptotically optimal algorithms.** We recall that, if $P_{GL}$ can be solved, then there exists a wide variety of algorithms that are asymptotically optimal, such as those presented by [7], [1], [5] and [22]. All these algorithms attempt to select arm $k \neq k^\star(\boldsymbol{\mu})$ a number of times $\eta_k^\star \ln T + o(\ln T)$ when $T \to \infty$, where $\boldsymbol{\eta}^\star$ is a solution to $P_{GL}$. The only requirement is that one is able to solve $P_{GL}$ several times in order to decide which arm to explore.

**Theorem 1** (Theorem 2 in [1]). *Consider Gaussian rewards with variance one and assume that for any $\boldsymbol{\mu} \in \mathcal{F}_{\leq m}$, a solution to the Graves-Lai problem can be computed. Then, the OSSB algorithm with parameters $\varepsilon = \gamma = 0$ is such that for all $\boldsymbol{\mu} \in \mathcal{F}_{\leq m}$, $\limsup_{T \to \infty} \frac{R(\boldsymbol{\mu}, T)}{\ln T} \leq C(m, \boldsymbol{\mu})$.*

For completeness, the pseudo-code of OSSB is provided in Appendix A.1. We now focus on how to solve $P_{GL}$ for multimodal bandit problems.

# 4 A computationally tractable algorithm to solve the Graves-Lai problem

In this section, we propose an algorithm to solve the Graves-Lai optimization problem in a computationally tractable manner, which constitutes our main contribution. The algorithm is rather intricate, and we go through its derivation step by step. The complete approach is summarized in Figure 1. For clarity, detailed proofs are provided in Appendix C.

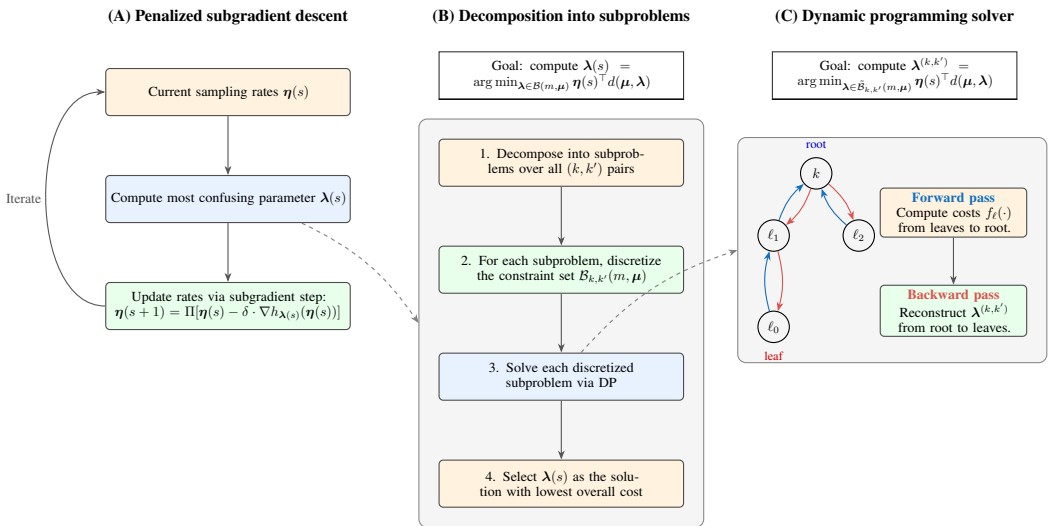

Figure 1: Summary of the procedure to solve $P_{GL}$.

## 4.1 Reducing the constraint of $P_{GL}$ to tractable subproblems

**On the difficulty of computing the constraint.** In the unimodal case where $m = 1$, solving $P_{GL}$ can be done in closed form, however for $m > 1$ this is much less straightforward. The main difficulty is to compute the value of the constraint $\inf_{\boldsymbol{\lambda} \in \mathcal{B}(m, \boldsymbol{\mu})} \boldsymbol{\eta}^\top d(\boldsymbol{\mu}, \boldsymbol{\lambda})$. While $\boldsymbol{\lambda} \mapsto \boldsymbol{\eta}^\top d(\boldsymbol{\mu}, \boldsymbol{\lambda})$ is usually a convex function, minimizing it over $\mathcal{B}(m, \boldsymbol{\mu})$ is difficult, due to the particular structure of the set $\mathcal{B}(m, \boldsymbol{\mu})$. Indeed, $\mathcal{B}(m, \boldsymbol{\mu})$ is not convex, nor is it connected. In Proposition 2 we show that, if one wanted to express $\mathcal{B}(m, \boldsymbol{\mu})$ as a union of $\mathcal{U}(K, m)$ convex sets (so that we could minimize $\boldsymbol{\eta}^\top d(\boldsymbol{\mu}, \boldsymbol{\lambda})$ over each set using convex programming), then one would require $\mathcal{U}(K, m)$ to be exponentially large in $K$. For example, if $G$ is a line graph with $K = 100$ nodes, and we consider multimodal functions with $m = 5$ modes, then the number of convex components $\mathcal{U}(K, m)$ must be greater than $10^5$.

**Proposition 2.** *Assume that $\mathcal{B}(m, \boldsymbol{\mu})$ can be written as a union of $\mathcal{U}(K, m)$ convex sets. Then for any $m > 1$, we have:*

$$\mathcal{U}(K, m) \geq \frac{((\deg(G) - 1)m)!}{(\deg(G)m)!}(K - (\deg(G) + 1)m)^m,$$

*hence $\mathcal{U}(K, m)$ grows exponentially with $m$.*

In contrast to the unimodal case ($m = 1$), where the most confusing parameter $\boldsymbol{\lambda}^\star$ is obtained by perturbing a single neighbor of the optimal arm $k^\star(\boldsymbol{\mu})$ (as shown by [2]), the multimodal setting ($m > 1$) introduces more complex possibilities. While one might expect the most confusing parameter to be such that $\lambda_k = \mu^\star$ for a single $k \in \mathcal{N}(\boldsymbol{\mu}) \setminus k^\star(\boldsymbol{\mu})$, and equal to $\boldsymbol{\mu}$ elsewhere, this intuition fails in general. Depending on the value of $\boldsymbol{\eta}$, it may be more confusing to set $\lambda_k = \mu^\star$ for $k \notin \mathcal{N}(\boldsymbol{\mu})$, and to ensure that $\boldsymbol{\lambda} \in \mathcal{F}_{\leq m}$, set $\lambda_{k'} = \lambda_\ell$ for some $k' \in \mathcal{M}(\boldsymbol{\mu}) \setminus k^\star(\boldsymbol{\mu})$ and $\ell$ such that $(k', \ell) \in E$. Figure 2 provides a concrete illustration of

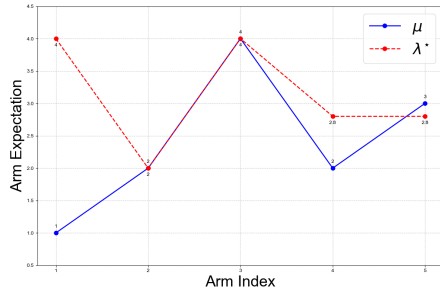

Figure 2: 2-modal example.

this phenomenon on a line graph, with $\boldsymbol{\lambda}^\star = \arg\min_{\boldsymbol{\lambda} \in \overline{\mathcal{B}}(2, \boldsymbol{\mu})} \boldsymbol{\eta}^\top d(\boldsymbol{\mu}, \boldsymbol{\lambda})$ for $\boldsymbol{\mu} = (1, 2, 4, 2, 3)$, $\boldsymbol{\eta} = (0.01, 0.25, 1, 0.25, 1)$, and Gaussian rewards with variance one.

**Restricting the constraint and solution spaces.** We first show some elementary properties of the solution, which will allow us to restrict both the spaces where $\boldsymbol{\eta}$ and $\boldsymbol{\lambda}$ lie. First, we decompose the constraint set as $\mathcal{B}(m, \boldsymbol{\mu}) = \cup_{k \neq k^\star(\boldsymbol{\mu})} \mathcal{B}_k(m, \boldsymbol{\mu})$ with $\mathcal{B}_k(m, \boldsymbol{\mu}) = \{\boldsymbol{\lambda} \in \mathcal{F}_{\leq m}, \lambda_{k^\star(\boldsymbol{\mu})} = \mu^\star, k^\star(\boldsymbol{\lambda}) = k\}$. Clearly, minimizing $\boldsymbol{\eta}^\top d(\boldsymbol{\mu}, \boldsymbol{\lambda})$ over $\boldsymbol{\lambda} \in \mathcal{B}(m, \boldsymbol{\mu})$ amounts to minimizing $\boldsymbol{\eta}^\top d(\boldsymbol{\mu}, \boldsymbol{\lambda})$ over $\boldsymbol{\lambda} \in \mathcal{B}_k(m, \boldsymbol{\mu})$ for each $k \neq k^\star(\boldsymbol{\mu})$. Proposition 3 states that this minimization problem is straightforward when $\boldsymbol{\mu}$ has strictly less than $m$ modes or when $k$ is in the neighborhood of the modes of $\boldsymbol{\mu}$.

**Proposition 3.** *Let $\boldsymbol{\eta} \geq 0$ and $k \neq k^\star(\boldsymbol{\mu})$. If $k \in \mathcal{N}(\boldsymbol{\mu})$ or $|\mathcal{M}(\boldsymbol{\mu})| < m$,*

$$\inf_{\boldsymbol{\lambda} \in \mathcal{B}_k(m, \boldsymbol{\mu})} \boldsymbol{\eta}^\top d(\boldsymbol{\mu}, \boldsymbol{\lambda}) = \eta_k d_k(\mu_k, \mu^\star),$$

*which is attained for $\boldsymbol{\lambda} = \boldsymbol{\mu} + (\mu^\star - \mu_k)e^{(k)} \in \overline{\mathcal{B}_k}(m, \boldsymbol{\mu})$.*

We now focus on the case $|\mathcal{M}(\boldsymbol{\mu})| = m$ and $k \notin \mathcal{N}(\boldsymbol{\mu})$. Proposition 4 shows that the constraint $\boldsymbol{\lambda} \in \mathcal{B}_k(m, \boldsymbol{\mu})$ is equivalent to a constraint on a compact set whose elements have entries comprised between the minimum and maximum of $\boldsymbol{\mu}$, and that have the same value $\mu^\star$ at $k$ and $k^\star(\boldsymbol{\mu})$.

**Proposition 4.** *Let $\boldsymbol{\eta} \geq 0$, $k \notin \mathcal{N}(\boldsymbol{\mu})$ and $\mathcal{B}_k'(m, \boldsymbol{\mu}) = \{\boldsymbol{\lambda} \in [\mu_\star, \mu^\star]^K \cap \mathcal{F}_{\leq m}, \lambda_k = \lambda_{k^\star(\boldsymbol{\mu})} = \mu^\star\}$. Then*

$$\inf_{\boldsymbol{\lambda} \in \mathcal{B}_k(m, \boldsymbol{\mu})} \boldsymbol{\eta}^\top d(\boldsymbol{\mu}, \boldsymbol{\lambda}) = \min_{\boldsymbol{\lambda} \in \mathcal{B}_k'(m, \boldsymbol{\mu})} \boldsymbol{\eta}^\top d(\boldsymbol{\mu}, \boldsymbol{\lambda}).$$

Proposition 5 shows that, in order to compute a solution to $P_{GL}$, we can restrict our attention to a compact region, and that the entries of $\boldsymbol{\eta}$ cannot be larger than $\mathfrak{B}(\boldsymbol{\mu})$. In turn, $\mathfrak{B}(\boldsymbol{\mu})$ may be interpreted as the regret predicted by the Lai-Robbins bound in absence of a multimodal structure, divided by the minimal gap.

**Proposition 5.** *There is a solution $\boldsymbol{\eta}^\star$ of $P_{GL}$ such that $\boldsymbol{\eta}^\star \in [0, \mathfrak{B}(\boldsymbol{\mu})]^K$ where $\mathfrak{B}(\boldsymbol{\mu}) = \frac{1}{\Delta_{\min}} \sum_{k:\Delta_k > 0} \frac{\Delta_k}{d_k(\mu_k, \mu^\star)}$.*

**Location of modes in subproblems.** We have shown in the previous section that the computation of the constraint of $P_{GL}$ can be reduced to solving the following subproblem for all $\boldsymbol{\eta} \geq 0$ and for each $k \in [K] \setminus \mathcal{N}(\boldsymbol{\mu})$:

$$\text{minimize}_{\boldsymbol{\lambda}} \ \boldsymbol{\eta}^\top d(\boldsymbol{\mu}, \boldsymbol{\lambda}) \text{ subject to } \boldsymbol{\lambda} \in \mathcal{B}_k'(m, \boldsymbol{\mu}). \qquad (P_{GL}(k))$$

We now present the most important structural result about the solution to $P_{GL}(k)$, which pertains to the location of its modes. Proposition 6 states that the modes of the solution to $P_{GL}(k)$ all lie in the set of modes of $\boldsymbol{\mu}$, apart from $k$, which must of course be a mode since it is a maximizer. This is in fact the reason why one is able to compute the solution to $P_{GL}(k)$. While this will be made clearer by exhibiting an efficient algorithm to compute the solution, it is understood searching over $\boldsymbol{\lambda}$ is much easier when the location of its modes is known.

**Proposition 6.** *Consider $\boldsymbol{\lambda}^\star$ the solution to $P_{GL}(k)$. Then $\mathcal{M}(\boldsymbol{\lambda}^\star) \subset \mathcal{M}(\boldsymbol{\mu}) \cup \{k\}$.*

If $|\mathcal{M}(\boldsymbol{\mu})| = m$, we have $|\mathcal{M}(\boldsymbol{\mu}) \cup \{k\}| = m + 1$, which implies that there must exist a mode $k'$ of $\boldsymbol{\mu}$ that is not a mode of $\boldsymbol{\lambda}^\star$, and all of the modes of $\boldsymbol{\lambda}^\star$ apart from $k$ are modes of $\boldsymbol{\mu}$. Additionally, we can assume that $k' \neq k^\star(\boldsymbol{\mu})$. Indeed, if $k' = k^\star(\boldsymbol{\mu})$, the constraint $\lambda_{k^\star(\boldsymbol{\mu})} = \mu^\star$ would yield $\lambda_\ell \geq \mu^\star$ for some neighbor $\ell$ of $k^\star(\boldsymbol{\mu})$. This cannot improve upon the solution given by Proposition 3. This means that we can restrict our attention to the sets $\mathcal{B}_{k,k'}(m, \boldsymbol{\mu})$ for $k' \in \mathcal{M}(\boldsymbol{\mu}) \setminus \{k^\star(\boldsymbol{\mu})\}$ with

$$\mathcal{B}_{k,k'}(m, \boldsymbol{\mu}) = \{\boldsymbol{\lambda} \in [\mu_\star, \mu^\star]^K, \mathcal{M}(\boldsymbol{\lambda}) \subset \mathcal{M}(\boldsymbol{\mu}) \cup \{k\} \setminus \{k'\}, \lambda_{k^\star(\boldsymbol{\mu})} = \lambda_k = \mu^\star\}$$

which is the set of vectors whose modes lie in $\mathcal{M}(\boldsymbol{\mu}) \cup \{k\} \setminus \{k'\}$ and attain their maximum at $k^\star(\boldsymbol{\mu})$ and $k$, and that have the same value as $\boldsymbol{\mu}$ at $k^\star(\boldsymbol{\mu})$. We must solve the subproblems, for $k \in [K] \setminus \mathcal{N}(\boldsymbol{\mu})$, $k' \in \mathcal{M}(\boldsymbol{\mu}) \setminus \{k^\star(\boldsymbol{\mu})\}$:

$$\text{minimize}_{\boldsymbol{\lambda}} \ \boldsymbol{\eta}^\top d(\boldsymbol{\mu}, \boldsymbol{\lambda}) \text{ subject to } \boldsymbol{\lambda} \in \mathcal{B}_{k,k'}(m, \boldsymbol{\mu}). \qquad (P_{GL}(k, k'))$$

**Discretizing the subproblems.** The last step before we can solve $P_{GL}(k, k')$ is to discretize the space in which $\boldsymbol{\lambda}$ lies, which will allow us to design a discrete search procedure over $\boldsymbol{\lambda}$. Proposition 7 states that discretizing each entry of $\boldsymbol{\lambda} \in \mathcal{B}_{k,k'}(m, \boldsymbol{\mu})$ with a grid of $n$ points $D(n, \boldsymbol{\mu})$, incurs a small approximation error that can be controlled, and vanishes at a rate inversely proportional to $n$. It is noted that this result is non trivial in the sense that there exists sets of large volume whose intersection with some grid can be empty, and holds because of the particular structure of $\mathcal{B}_{k,k'}(m, \boldsymbol{\mu})$. In essence, we can round $\boldsymbol{\lambda}$ to ensure both a small rounding error while leaving the set of modes of $\boldsymbol{\lambda}$ unchanged.

**Proposition 7.** *Consider $D(n, \boldsymbol{\mu})$ the following uniform discretization of $[\mu_\star, \mu^\star]$ with $n$ discretization points:*

$$D(n, \boldsymbol{\mu}) = \{\mu_\star + (i/n)(\mu^\star - \mu_\star), i \in [n]\}.$$

*Then there exists $\tilde{\boldsymbol{\lambda}} \in \mathcal{B}_{k,k'}(m, \boldsymbol{\mu}) \cap D(n, \boldsymbol{\mu})^K$ such that for any $\boldsymbol{\eta} \in [0, \mathfrak{B}(\boldsymbol{\mu})]^K$:*

$$\boldsymbol{\eta}^\top d(\boldsymbol{\mu}, \tilde{\boldsymbol{\lambda}}) - \frac{\mathfrak{C}(\boldsymbol{\mu})}{n} \leq \min_{\boldsymbol{\lambda} \in \mathcal{B}_{k,k'}(m, \boldsymbol{\mu})} \boldsymbol{\eta}^\top d(\boldsymbol{\mu}, \boldsymbol{\lambda}) \leq \boldsymbol{\eta}^\top d(\boldsymbol{\mu}, \tilde{\boldsymbol{\lambda}})$$

*with $\mathfrak{C}(\boldsymbol{\mu}) = \mathrm{diam}(G)(\mu^\star - \mu_\star)\mathfrak{A}(\boldsymbol{\mu})\mathfrak{B}(\boldsymbol{\mu})K$.*

### 4.2 Computing the constraint sets via dynamic programming

We now explain how to efficiently solve the discretized version of $P_{GL}(k, k')$, namely

$$\text{minimize}_{\boldsymbol{\lambda}} \ \boldsymbol{\eta}^\top d(\boldsymbol{\mu}, \boldsymbol{\lambda}) \text{ subject to } \boldsymbol{\lambda} \in \tilde{B}_{k,k'}(m, \boldsymbol{\mu}) \qquad (\tilde{P}_{GL}(k, k'))$$

for $\tilde{B}_{k,k'}(m, \boldsymbol{\mu}) = \mathcal{B}_{k,k'}(m, \boldsymbol{\mu}) \cap D(n, \boldsymbol{\mu})^K$. We use a dynamic programming procedure which necessitates viewing $G$ as a directed tree $G^k$, obtained by performing depth-first search on the undirected tree $G$ starting at node $k$ (which is consequently the root of $G^k$). We recall the notations $\mathcal{C}(\ell), \mathcal{D}(\ell), p(\ell)$ and $p^2(\ell)$ to denote the children, descendants, parent and grandparent of a node $\ell$ in $G^k$. The high-level idea of the procedure is to compute the value of $\tilde{P}_{GL}(k, k')$ recursively with a formula that relates the minimal obtainable value of $\sum_{j \in \mathcal{D}(\ell) \cup \{\ell\}} \eta_j d_j(\mu_j, \lambda_j)$ to that of $\sum_{j \in \mathcal{D}(\ell)} \eta_j d_j(\mu_j, \lambda_j)$ for any node $\ell$. Note that when $\ell = k$, the former is equal to the value of $\tilde{P}_{GL}(k, k')$, and when $\ell$ is a leaf of $G^k$, the latter is equal to 0. This recursion formula heavily relies on the fact that all modes of the solution to $\tilde{P}_{GL}(k, k')$ are in $\mathcal{M}(\boldsymbol{\mu}) \cup \{k\} \setminus \{k'\}$.

We now introduce some important quantities for our dynamic programming approach. For a node $\ell \neq k$ in $G^k$, we define $f_\ell(z, u)$ as the minimal obtainable value of

$$\sum_{j \in \mathcal{D}(\ell) \cup \{\ell\}} \eta_j d_j(\mu_j, \lambda_j)$$

subject to the constraints $\boldsymbol{\lambda} \in \tilde{B}_{k,k'}(m, \boldsymbol{\mu})$, $\lambda_\ell = z$ and $\lambda_\ell > \lambda_{p(\ell)}$ if $u = 1$ (resp. $\lambda_\ell \leq \lambda_{p(\ell)}$ if $u = -1$). To simplify the notations further, we introduce the following auxiliary functions[2]:

$$f_\ell^\star(z, +1) = \min_{w > z} f_\ell(w, +1), \quad f_\ell^\star(z, -1) = \min_{w \leq z} f_\ell(w, -1), \quad f_\ell^\diamond(z) = \min_{u \in \{-1, +1\}} f_\ell^\star(z, u),$$

which represent the minimal obtainable value of $\sum_{j \in \mathcal{D}(\ell) \cup \{\ell\}} \eta_j d_j(\mu_j, \lambda_j)$ for $\boldsymbol{\lambda} \in \tilde{B}_{k,k'}(m, \boldsymbol{\mu})$ when $\lambda_\ell > \lambda_{p(\ell)} = z$, $\lambda_\ell \leq \lambda_{p(\ell)} = z$ and $\lambda_{p(\ell)} = z$, respectively. Finally, to ensure that the constraint $\lambda_{k^\star(\boldsymbol{\mu})} = \mu^\star$ is satisfied during the dynamic programming procedure, we set [3] $\eta_{k^\star(\boldsymbol{\mu})} = +\infty$, and we use the convention that $\eta_{k^\star(\boldsymbol{\mu})} d_{k^\star(\boldsymbol{\mu})}(\mu^\star, z)$ equals 0 if $z = \mu^\star$ and $+\infty$ otherwise.

**Proposition 8.** *The functions $f_\ell(z, u)$ for $\ell \in [K]$, $z \in D(n, \boldsymbol{\mu})$ and $u \in \{-1, +1\}$ obey the following recursion:*

*If $\ell \in \mathcal{M}(\boldsymbol{\mu}) \cup \{k\} \setminus \{k'\}$: $f_\ell(z, u) = \eta_\ell d_\ell(\mu_\ell, z) + \sum_{j \in \mathcal{C}(\ell)} f_j^\diamond(z)$, and if $\ell \notin \mathcal{M}(\boldsymbol{\mu}) \cup \{k\} \setminus \{k'\}$:*

$$f_\ell(z, u) = \begin{cases} \eta_\ell d_\ell(\mu_\ell, z) + \sum_{j \in \mathcal{C}(\ell)} f_j^\diamond(z) & \text{if } u = -1 \\ \eta_\ell d_\ell(\mu_\ell, z) + \sum_{j \in \mathcal{C}(\ell)} f_j^\diamond(z) + \min_{v \in \mathcal{C}(\ell)} g_v(z) & \text{if } u = +1, \mathcal{C}(\ell) \neq \emptyset \\ +\infty & \text{if } u = +1, \mathcal{C}(\ell) = \emptyset \end{cases}$$

---

[2]The minima are taken with the implicit constraint $w \in D(n, \boldsymbol{\mu})$.

[3]Recall that the value of $\eta_{k^\star(\boldsymbol{\mu})}$ has no impact on the solution to $\tilde{P}_{GL}(k, k')$.

*where $g_v(z) = \min\{f_v^\star(z, +1), f_v(z, -1)\} - f_v^\diamond(z)$, and the value of $\tilde{P}_{GL}(k, k')$ equals $f_k(\mu^\star, u)$ for any $u \in \{-1, +1\}$.*

Since the discretized search space $D(n, \boldsymbol{\mu})$ is finite, we can straightforwardly compute the values of $f_\ell(z, u), f_\ell^\star(z, u)$ and $f_\ell^\diamond(z)$ for each $\ell \in [K]$, $z \in D(n, \boldsymbol{\mu})$ and $u \in \{-1, +1\}$ with the dynamic programming equations of Proposition 8. The solution $\boldsymbol{\lambda}^\star$ of $\tilde{P}_{GL}(k, k')$ can then be obtained recursively as in Corollary 1, in which the condition $\ell = \arg\min_{v \in \mathcal{C}(p(\ell))} g_v(\lambda_{p(\ell)}^\star)$ can be understood as $\ell$ being the children of $p(\ell)$ that induces the smallest cost when constrained by the value of $p(\ell)$.

**Corollary 1.** *The solution $\boldsymbol{\lambda}^\star$ of $\tilde{P}_{GL}(k, k')$ is such that $\lambda_k^\star = \mu^\star$ and for $\ell \neq k$:*

*If $p(\ell) \notin \mathcal{M}(\boldsymbol{\mu}) \cup \{k\} \setminus \{k'\}$, $\lambda_{p(\ell)}^\star > \lambda_{p^2(\ell)}^\star$ and $\ell = \arg\min_{v \in \mathcal{C}(p(\ell))} g_v(\lambda_{p(\ell)}^\star)$:*

$$\lambda_\ell^\star = \begin{cases} \arg\min_{z > \lambda_{p(\ell)}^\star} f_\ell(z, +1) & \text{if } f_\ell^\star(\lambda_{p(\ell)}^\star, +1) \leq f_\ell(\lambda_{p(\ell)}^\star, -1) \\ \lambda_{p(\ell)}^\star & \text{if } f_\ell^\star(\lambda_{p(\ell)}^\star, +1) > f_\ell(\lambda_{p(\ell)}^\star, -1). \end{cases}$$

*and otherwise* $\lambda_\ell^\star = \begin{cases} \arg\min_{z > \lambda_{p(\ell)}^\star} f_\ell(z, +1) & \text{if } f_\ell^\star(\lambda_{p(\ell)}^\star, +1) \leq f_\ell^\star(\lambda_{p(\ell)}^\star, -1) \\ \arg\min_{z \leq \lambda_{p(\ell)}^\star} f_\ell(z, -1) & \text{if } f_\ell^\star(\lambda_{p(\ell)}^\star, +1) > f_\ell^\star(\lambda_{p(\ell)}^\star, -1). \end{cases}$

*Furthermore, computing the solution to $\tilde{P}_{GL}(k, k')$ using the procedure of Proposition 8 and Corollary 1 can be done in time and memory $O(nK)$.*

**Overall time complexity.** This procedure allows us to solve the subproblems $\tilde{P}_{GL}(k, k')$ for all $k \notin \mathcal{N}(\boldsymbol{\mu})$ and $k' \in \mathcal{M}(\boldsymbol{\mu}) \setminus \{k^\star(\boldsymbol{\mu})\}$. By comparing these solutions with the trivial parameters from Proposition 3, we can find the most confusing parameter in $\overline{\mathcal{B}}(m, \boldsymbol{\mu}) \cap D(n, \boldsymbol{\mu})^K$ for any sampling rate $\boldsymbol{\eta}$ in time $O(K^2 mn)$. In practice, these subproblems are independent and can be solved in parallel. In Appendix E, we describe a more involved dynamic program that runs in time $O(Kn)$ without requiring parallelism.

**Illustration of the dynamic programming approach.** We now illustrate the computation of the most confusing parameter in $\overline{\mathcal{B}}(m, \boldsymbol{\mu}) \cap D(n, \boldsymbol{\mu})^K$ with the line graph example of Figure 2. There, the divergence is $d_k(\lambda_k, \mu_k) = \frac{1}{2}(\lambda_k - \mu_k)^2$, the optimal arm is $k^\star(\boldsymbol{\mu}) = 3$, the modes are $\mathcal{M}(\boldsymbol{\mu}) = \{3, 5\}$, and their neighborhood is $\mathcal{N}(\boldsymbol{\mu}) = \{2, 3, 4, 5\}$. If $k = k^\star(\boldsymbol{\lambda})$ is chosen in $\mathcal{N}(\boldsymbol{\mu})$, applying Proposition 3 yields

$$\inf_{\boldsymbol{\lambda} \in \mathcal{B}_k(m, \boldsymbol{\mu})} \boldsymbol{\eta}^\top d(\boldsymbol{\mu}, \boldsymbol{\lambda}) = \frac{1}{2}.$$

Otherwise, we must have $k = 1$, and the only choice for the mode of $\boldsymbol{\mu}$ to be removed is $k' = 5$.

We can then solve $\tilde{P}_{GL}(k, k')$ for $(k, k') = (1, 5)$ by applying Proposition 8 as follows. We first form the directed tree $G^1$ as a line graph, rooted at 1, with leaf node 5. For each node $\ell$ and each grid value $z \in D(n, \mu)$ the program computes and stores in memory the values

$$f_\ell(z, -1), \qquad f_\ell(z, +1),$$

together with the auxiliary minima $f_\ell^\star(z, \cdot)$ and $f_\ell^\diamond(z)$, as defined in Proposition 8. The leaf entry is obtained directly from the divergence:

$$f_5(z, -1) = \frac{\eta_5}{2}(\mu_5 - z)^2, \qquad f_5(z, +1) = +\infty,$$

and $f_5^\star(z, \cdot), f_5^\diamond(z)$ are then computed by minimizing over the grid $D(n, \mu)$. For an internal node $\ell \neq 5$, the recursion in Proposition 8 is applied: each entry $f_\ell(z, \cdot)$ is derived from the already computed child cost values as prescribed in the proposition. Finally, the value of the discretized subproblem $\tilde{P}_{GL}(k, k')$ is read off at the root as $f_1(\mu^\star, +1)$, where $\mu^\star = 4$ in the present example. The minimizer is finally recovered by backtracking, as described in Corollary 1. In the limit $n \to \infty$, this minimizer approaches $\boldsymbol{\lambda}^\star = (4, 2, 4, 2.8, 2.8)$. This non-trivial confusing parameter turns out to be the global minimizer of $P_{GL}$ as its value approaches $0.145 < 0.5$.

### 4.3 Solving $P_{GL}$ via penalized subgradient descent

We are now capable of computing a minimizer of $\boldsymbol{\eta}^\top d(\boldsymbol{\mu}, \boldsymbol{\lambda})$ over $\overline{\mathcal{B}}(m, \boldsymbol{\mu}) \cap D(n, \boldsymbol{\mu})^K$, the constraint in the original problem $P_{GL}$, with discretization. The last step to close the loop is to use an iterative procedure to derive an approximate solution to $P_{GL}$. The simplest way to understand this procedure is to view it as a projected subgradient descent for the convex function

$$h : \boldsymbol{\eta} \mapsto \boldsymbol{\eta}^\top \boldsymbol{\Delta} + \gamma \max \left[ 1 - \min_{\boldsymbol{\lambda} \in \overline{\mathcal{B}}(m, \boldsymbol{\mu}) \cap D(n, \boldsymbol{\mu})^K} \{ \boldsymbol{\eta}^\top d(\boldsymbol{\mu}, \boldsymbol{\lambda}) \}, 0 \right]$$

which can be interpreted as the objective of $P_{GL}$ with a discretized constraint space and where the hard constraints have been replaced by a penalty similar to the hinge loss function, and where $\gamma$ controls the magnitude of the penalty. The projection step is used to enforce the constraint $\boldsymbol{\eta} \geq 0$. We show in Proposition 9 that the minimizer of $h(\boldsymbol{\eta})$ subject to the constraint $\boldsymbol{\eta} \geq 0$ is an approximate solution to $P_{GL}$.

**Proposition 9.** *Consider a step size* $\delta^2 = \frac{K\mathfrak{B}(\boldsymbol{\mu})^2}{t\mathfrak{E}(\boldsymbol{\mu})^2}$ *where $t$ is the number of iterations,* $\mathfrak{E}(\boldsymbol{\mu}) = \|\boldsymbol{\Delta}\|_2 + \gamma K^{3/2}\mathfrak{A}(\boldsymbol{\mu})(\mu^\star - \mu_\star)$, *a penalty* $\gamma = 2\max_{k, \Delta_k > 0} \frac{\Delta_k}{d(\mu_k, \mu^\star)}$ *and an iterative procedure* $\boldsymbol{\eta}(1) = 0$, *and for $s < t$:*

$$\boldsymbol{\eta}(s+1) = \Pi \left[ \boldsymbol{\eta}(s) - \delta \left( \boldsymbol{\Delta} - \gamma d(\boldsymbol{\mu}, \boldsymbol{\lambda}(s)) \mathbb{1}\{ \boldsymbol{\eta}^\top d(\boldsymbol{\mu}, \boldsymbol{\lambda}(s)) < 1 \} \right) \right]$$

*where* $\boldsymbol{\lambda}(s) \in \arg\min_{\boldsymbol{\lambda} \in \overline{\mathcal{B}}(m, \boldsymbol{\mu}) \cap D(n, \boldsymbol{\mu})^K} \boldsymbol{\eta}(s)^\top d(\boldsymbol{\mu}, \boldsymbol{\lambda})$ *and* $\Pi[x] = (\max(x_k, 0))_{k \in [K]}$ *is the projection on the positive orthant. Define* $\bar{\boldsymbol{\eta}}(t) = (1/t) \sum_{s=1}^t \boldsymbol{\eta}(s)$ *the average iterate and a scaled version* $\tilde{\boldsymbol{\eta}}(t) = \bar{\boldsymbol{\eta}}(t) \left( 1 - \frac{\mathfrak{C}(\boldsymbol{\mu})}{n} - 2\frac{\mathfrak{F}(\boldsymbol{\mu})}{\gamma\sqrt{t}} \right)^{-1}$ *for* $\mathfrak{F}(\boldsymbol{\mu}) = \sqrt{K}\mathfrak{B}(\boldsymbol{\mu})\mathfrak{E}(\boldsymbol{\mu})$. *Then* $\tilde{\boldsymbol{\eta}}(t)$ *is a feasible solution to $P_{GL}$ with value at most:*

$$\tilde{\boldsymbol{\eta}}(t)^\top \boldsymbol{\Delta} \leq \left( 1 - \frac{\mathfrak{C}(\boldsymbol{\mu})}{n} - 2\frac{\mathfrak{F}(\boldsymbol{\mu})}{\gamma\sqrt{t}} \right)^{-1} \left( C(m, \boldsymbol{\mu}) + \frac{\mathfrak{F}(\boldsymbol{\mu})}{\sqrt{t}} \right) \quad if \quad \frac{\mathfrak{C}(\boldsymbol{\mu})}{n} - 2\frac{\mathfrak{F}(\boldsymbol{\mu})}{\gamma\sqrt{t}} < 1.$$

**Putting it all together.** We end this section by stating our main result, which is a direct consequence of the previous propositions, and propose a computationally tractable algorithm in order to compute an approximate solution to $P_{GL}$. With the more intricate dynamic programming scheme presented in Appendix E, its time complexity can be improved to $O(Knt)$.

**Theorem 2.** *Consider the algorithm which outputs $\tilde{\boldsymbol{\eta}}(t)$ after $t$ iterations of the scheme described in Proposition 9 with $n$ discretization points. At each step $s \leq t$, $\boldsymbol{\lambda}(s)$ is computed by solving $\tilde{P}_{GL}(k, k')$ for all $k \notin \mathcal{N}(\boldsymbol{\mu})$ and all $k' \in \mathcal{M}(\boldsymbol{\mu}) \setminus \{k^\star(\boldsymbol{\mu})\}$ using the dynamic programming scheme of Proposition 8. This algorithm runs in time $O(K^2 mnt)$ and space $O(Knt)$ and yields $\tilde{\boldsymbol{\eta}}(t)$, a feasible solution to $P_{GL}$ with value at most:*

$$\tilde{\boldsymbol{\eta}}(t)^\top \boldsymbol{\Delta} \leq \left( 1 - \frac{\mathfrak{C}(\boldsymbol{\mu})}{n} - 2\frac{\mathfrak{F}(\boldsymbol{\mu})}{\gamma\sqrt{t}} \right)^{-1} \left( C(m, \boldsymbol{\mu}) + \frac{\mathfrak{F}(\boldsymbol{\mu})}{\sqrt{t}} \right) \quad if \quad \frac{\mathfrak{C}(\boldsymbol{\mu})}{n} - 2\frac{\mathfrak{F}(\boldsymbol{\mu})}{\gamma\sqrt{t}} < 1.$$

The parameters $n$ and $t$ can be chosen by the learner. Since the time complexity grows linearly in $nt$, and the optimization error is proportional to $1/n + 1/\sqrt{t}$, given a time budget constraint $nt = a$, one may choose $n = a^{1/3}$ and $t = a^{2/3}$, which yields an optimization error proportional to $a^{-1/3}$.

## 5 Local search strategies and peaked functions

In this section, we consider algorithms that primarily explore arms in $\mathcal{N}(\boldsymbol{\mu})$, where we recall that $\mathcal{N}(\boldsymbol{\mu})$ is the set of all modes of $\boldsymbol{\mu}$ and their neighbors. The proofs are deferred to Appendix D.

**Local search.** To analyze such algorithms within the Graves-Lai framework, we connect this behavior to the properties of the corresponding sampling rate vector $\boldsymbol{\eta}$. Let $N_k(t) = \sum_{s=1}^t \mathbb{1}\{k(s) = k\}$ denote the number of times arm $k$ has been selected by up to round $t$. The asymptotic sampling rate for arm $k \in [K]$ is given by $\eta_k = \limsup_{T \to \infty} \frac{\mathbb{E}[N_k(T)]}{\ln T}$. An algorithm performs local search if $\eta_k = 0$ for all $k \notin \mathcal{N}(\boldsymbol{\mu})$. This leads directly to the following definition.

**Definition 1.** *Consider $\boldsymbol{\eta} \in (\mathbb{R}^+)^K$ a feasible solution to $P_{GL}$. We say that $\boldsymbol{\eta}$ is a local search strategy if and only if $\eta_k = 0$ for all $k \notin \mathcal{N}(\boldsymbol{\mu})$. Further define $C_{\mathrm{loc}}(m, \boldsymbol{\mu})$ the optimal value of $P_{GL}$ restricted to the set of local search strategies.*

The appeal of local search strategies stems from two key properties: they are provably optimal in the unimodal case ($m = 1$), and are conceptually simpler than non-local strategies, which may explore all arms.

**Suboptimality of local search strategies.** Unfortunately, and rather counterintuitively, not only are local search strategies suboptimal, the performance gap between local and non-local strategies is not upper bounded. More precisely, the ratio between the value of the best local strategy and the value of the best strategy can be arbitrarily large. Local search strategies are suboptimal because for every mode $k \neq k^\star(\boldsymbol{\mu})$ and every neighbor $\ell$ of $k$, they must be able to check that $\mu_k > \mu_\ell$, requiring a number of samples proportional to $\frac{1}{d_\ell(\mu_\ell, \mu_k)}$, which can cause an arbitrarily large amount of regret if the function is *flat* in the neighborhood of $k$, so that $\mu_k$ is very close to $\mu_\ell$.

**Theorem 3.** *Assume that $|\mathcal{N}(\boldsymbol{\mu})| < K$. Then the following bounds hold:*

$$C_{\mathrm{loc}}(m, \boldsymbol{\mu}) \geq \sum_{k \in \mathcal{M}(\boldsymbol{\mu}) \setminus \{k^\star(\boldsymbol{\mu})\}} \frac{\Delta_k}{d_k(\mu_k, \mu_k - \delta_k)} \quad and \quad C(m, \boldsymbol{\mu}) \leq \sum_{k \in [K]} \frac{\Delta_k}{d_k(\mu_k, \mu^\star)}$$

*where $\delta_k = \min_{\ell:(k,\ell) \in E} |\mu_k - \mu_\ell| > 0$. As a consequence, $\sup_{\boldsymbol{\mu} \in \mathcal{F}_m} \frac{C_{\mathrm{loc}}(m, \boldsymbol{\mu})}{C(m, \boldsymbol{\mu})} = +\infty$, i.e., the performance ratio between local and non-local strategies is unbounded.*

In particular, this result implies that the `IMED-MB` algorithm from [18], which uses a local search strategy, cannot be asymptotically optimal, which contradicts the statement of their Theorem 2. This is rigorously shown in Appendix D.2.

**Peaked reward functions.** While in general local search strategies can be far from optimal, there exists a smaller subclass of reward functions on which they can be shown to be quasi-optimal, up to a constant multiplicative factor. We call these reward functions *peaked* in the sense that they are not flat around their modes.

**Definition 2.** *A reward function $\boldsymbol{\mu} \in \mathbb{R}^K$ is $\kappa$-peaked if and only if for all $k \in \mathcal{M}(\boldsymbol{\mu})$ and all $\ell$ neighbor of $k$ we have $d_k(\mu_k, \mu^\star) \leq \kappa d_k(\mu_k, \mu_k - \delta_k/2)$ and $d_\ell(\mu_\ell, \mu^\star) \leq \kappa d_\ell(\mu_\ell, \mu_k - \delta_k/2)$.*

In particular, when rewards are Gaussian, the condition above reduces to a simpler one: for all modes $k$, the gap between $k$ and its neighbors should be at least proportional to the gap between $k$ and the optimal arm, so that indeed, the function cannot be *flat* around the modes.

**Proposition 10.** *Consider Gaussian rewards with fixed variance. Then $\boldsymbol{\mu}$ is $\kappa$-peaked if and only if for all $k \in \mathcal{M}(\boldsymbol{\mu})$ we have $\Delta_k \leq \delta_k(\frac{\sqrt{\kappa}}{2} - 1)$.*

Proposition 11 shows that for $\kappa$-peaked reward functions, there exists local search strategies that are a factor $\kappa$ from optimal.

**Proposition 11.** *Assume that $\boldsymbol{\mu} \in \mathcal{F}_m$. Then the following bounds hold:*

$$C_{\mathrm{loc}}(m, \boldsymbol{\mu}) \leq \sum_{k \in \mathcal{M}(\boldsymbol{\mu}) \setminus \{k^\star(\boldsymbol{\mu})\}} \frac{\Delta_k}{d_k(\mu_k, \mu_k - \delta_k/2)} + \sum_{k \in \mathcal{M}(\boldsymbol{\mu})} \sum_{\ell:(k,\ell) \in E} \frac{\Delta_\ell}{d_\ell(\mu_\ell, \mu_k - \delta_k/2)}$$

$$C(m, \boldsymbol{\mu}) \geq \sum_{k \in \mathcal{N}(\boldsymbol{\mu}) \setminus \{k^\star(\boldsymbol{\mu})\}} \frac{\Delta_k}{d_k(\mu_k, \mu^\star)}$$

*and by corollary, if $\boldsymbol{\mu}$ is $\kappa$-peaked, there exists local search strategies within a constant factor:* $\sup_{\boldsymbol{\mu} \in \mathcal{F}_m, \kappa-\text{peaked}} \frac{C_{\mathrm{loc}}(m, \boldsymbol{\mu})}{C(m, \boldsymbol{\mu})} \leq \kappa$.

# 6 Numerical experiments

In this section, we conduct numerical experiments to demonstrate the benefit of properly exploiting the multimodal structure. To this end, we implement `OSSB` from [1] and use our approach to solve

$P_{GL}$. At round $t$, OSSB samples as dictated by the solution $\boldsymbol{\eta}^\star(t)$ to the Graves-Lai problem for the empirical estimate $\hat{\boldsymbol{\mu}}(t)$ of $\boldsymbol{\mu}$, which is given by $\hat{\mu}_k(t) = \frac{\sum_{s=1}^t X_{k,s} \mathbb{1}\{k(s)=k\}}{\max(1, N_k(t))}$. We compare the cumulative regret of two algorithms:

(i) *Multimodal OSSB*: the OSSB algorithm where the Graves-Lai problem is solved using our proposed method,

(ii) *Classical OSSB*: the OSSB algorithm for unstructured bandits, which serves as a baseline.

The pseudo-code of OSSB and further details regarding the experiment are deferred to Appendix A.1.

$G$ is chosen as a fixed binary tree of height two (resulting in $K = 7$ arms). We consider instances $\boldsymbol{\mu} \in \mathcal{F}_2$ with rewards from arm $k \in [K]$ drawn from a Gaussian distribution $\mathcal{N}(\mu_k, 1)$. The mean rewards $\mu_k$ are generated as a sum of exponential functions centered on the modes in $\mathcal{M}(\boldsymbol{\mu})$:

$$\mu_k = \sum_{j \in \mathcal{M}(\boldsymbol{\mu})} \left(1 + \mathbb{1}_{j=k^\star(\boldsymbol{\mu})}\right) \exp\left(-\rho_{jk}/\sigma\right),$$

where $\rho_{jk}$ is the shortest path distance between nodes $j$ and $k$ in $G$. We choose $\mathcal{M}(\boldsymbol{\mu}) = \{4, 6\}$ and $k^\star(\boldsymbol{\mu}) = 6$. The parameter $\sigma$ controls how peaked the reward function is: a small $\sigma$ leads to sharp peaks with modes well-separated from their neighbors, whereas a large $\sigma$ creates flatter modes.

We consider two instances: $\sigma = 0.5$ (easy instance) and $\sigma = 4$ (hard instance). We run the experiment up to a horizon of $T = 10{,}000$. To reduce the computational burden of solving $P_{GL}$ at each round, we only update $\boldsymbol{\eta}^\star(t)$ when $t = 2^k$ for $k \in \{0, \ldots, \lfloor \log_2 T \rfloor\}$. The cumulative regret is averaged over $500$ trials. The results are presented in Figure 3, where the shaded regions have radius one standard error. In both settings, multimodal OSSB exhibits superior performance over classical OSSB.

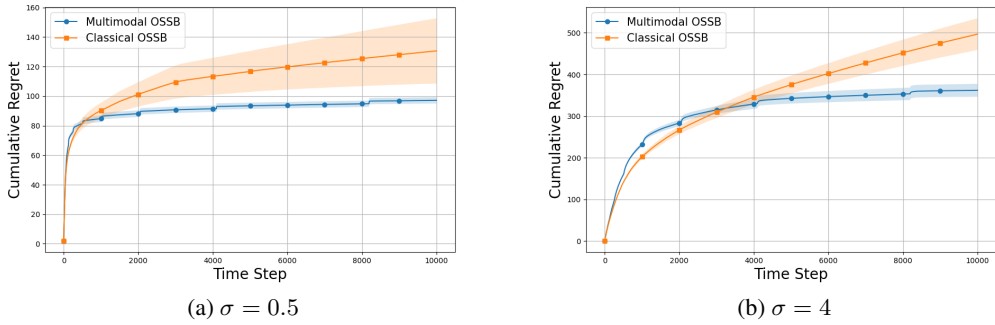

(a) $\sigma = 0.5$          (b) $\sigma = 4$

Figure 3: Cumulative regret as a function of the number of rounds.

Further experiments on the runtime of our dynamic programming approach are deferred to Appendices A.2 and E.8. The code used for the experiments is available at `https://github.com/wilrev/MultimodalBandits`.

# 7 Conclusion

We have considered a stochastic multi-armed bandit with i.i.d. rewards and a multimodal reward structure, and have proposed the first known computationally tractable algorithm to solve the Graves-Lai optimization problem, which we have shown is a requirement to implement asymptotically optimal algorithms, as the performance ratio between local and non-local strategies can be arbitrarily large. We believe that an interesting direction for future work is to characterize the minimal computational complexity necessary to solve this problem in terms of the number of arms, modes and graph structure.

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

# A    Experimental details

## A.1    OSSB for multimodal bandits

As explained in Section 3, asymptotically optimal algorithms for structured bandits attempt to sample $k \neq k^\star(\boldsymbol{\mu})$ a number of times $\eta_k^\star \ln T + o(\ln T)$ when $T \to \infty$, where $\boldsymbol{\eta}^\star$ is a solution to $P_{GL}$. The OSSB algorithm of [1] does so by sampling as dictated by the solution to the Graves-Lai problem for the empirical estimate $\hat{\boldsymbol{\mu}}(t)$ of $\boldsymbol{\mu}$, updated at each round $t$. The pseudo-code of OSSB for multimodal bandits is given in Algorithm 1.

---

**Algorithm 1** Optimal Sampling for Structured Bandits (OSSB)

---

$N_k(1) \leftarrow 0, \hat{\mu}_k(1) \leftarrow 0 \ \forall k \in [K]$ {Initialization}
**for** $t = 1, \ldots, T$ **do**
    Compute $\boldsymbol{\eta}^\star(t)$ the solution to $P_{GL}$ for $\hat{\boldsymbol{\mu}}(t)$
    **if** $\forall k \in [K], N_k(t) \geq \eta_k^\star(t) \ln t$ **then**
        $k(t) \leftarrow \arg \max_{k \in [K]} \hat{\mu}_k(t)$ {Exploitation, ties broken arbitrarily}
    **else**
        $k(t) \leftarrow \arg \min_{k \in [K]} \dfrac{N_k(t)}{\eta_k^\star(\hat{\boldsymbol{\mu}}(t))}$ {Exploration, ties broken arbitrarily}
    **end if**
    Observe $X_{k,(t),t}$
    **for** $k \neq k(t)$ **do**
        $\hat{\mu}_k(t+1) \leftarrow \hat{\mu}_k(t)$
        $N_k(t+1) \leftarrow N_k(t)$
    **end for**
    $\hat{\mu}_{k(t)}(t+1) \leftarrow \dfrac{X_{k(t),t} + \hat{\mu}_{k(t)}(t) N_{k(t)}(t)}{N_{k(t)}(t)+1}$
    $N_{k(t)}(t+1) \leftarrow N_{k(t)}(t) + 1$
**end for**

---

We implement two versions of OSSB. Each version is associated with a specific sampling strategy $\boldsymbol{\eta}^\star(t)$, which it aims to follow at round $t$. Let $\hat{\boldsymbol{\Delta}}_k(t) = \max_{k' \in [K]} \hat{\mu}_{k'}(t) - \hat{\mu}_k(t)$. These strategies are given by:

- the solution to $P_{GL}$ for $\hat{\boldsymbol{\mu}}(t)$, as computed by our algorithm, with the convention $\eta_k^\star(t) = 0$ if $k \in \arg \max \hat{\boldsymbol{\mu}}(t)$ (*Multimodal OSSB, Algorithm 1*)

- $\eta_k^\star(t) = \frac{1}{d_k(\hat{\mu}_k(t), \hat{\boldsymbol{\mu}}(t)^\star)} \mathbb{1}\{\hat{\boldsymbol{\Delta}}_k(t) > 0\}$ (*Classical OSSB, rates given by the Lai-Robbins bound [13]*).

The specific instances $\boldsymbol{\mu} \in \mathcal{F}_2$ considered in the experiment of Section 6 are shown in Figure 4.

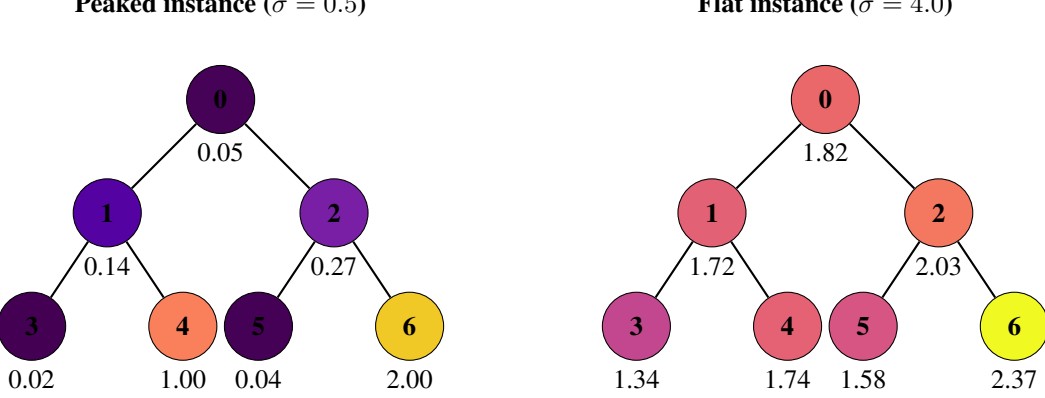

Figure 4: 2-modal reward instances on the binary tree $G$ ($K = 7$, $\mathcal{M}(\boldsymbol{\mu}) = \{4, 6\}$, $k^\star(\boldsymbol{\mu}) = 6$).

In this specific experiment, instead of the penalized subgradient subroutine described in Proposition 9, we used the Sequential Least SQuares Programming (SLSQP) method from [12] and implemented in SciPy by [23] for faster convergence towards a solution to $P_{GL}$ for each $\hat{\boldsymbol{\mu}}(t)$, and we only solve $P_{GL}$ when $t = 2^k$ for $k \in \{0, \ldots, \lfloor \log_2 T \rfloor\}$. We used $n = 100$ discretization points.

## A.2    Runtime experiment

In this experiment, we evaluate the runtime of the algorithm of Theorem 2 with respect to the number of arms $K$. We generate line graphs with varying numbers of arms $K \in \{20, 25, 30, \ldots, 70\}$, number of modes $|M(\boldsymbol{\mu})| = m \in \{2, 3, 4, 5\}$, for $n = 100$ discretization points and $t = 100$ iterations of penalized subgradient descent. The reward instances $\boldsymbol{\mu}$ are generated as in Section 6 with $\sigma = 2$. To ensure that they are always $m$-modal, the position of $k^\star(\boldsymbol{\mu})$ and of the other modes of $\boldsymbol{\mu}$ are chosen to be as spread out as possible. We perform 5 trials per configuration $(K, m)$. Figure 5 displays the average runtime as a function of $K$ for each value of $m$ on a log-log plot, as well as the corresponding slopes obtained from log-log regression. The runtime exhibits an approximately quadratic growth with the number of arms, which aligns with the time complexity $O(K^2 mnt)$ stated in Theorem 2.

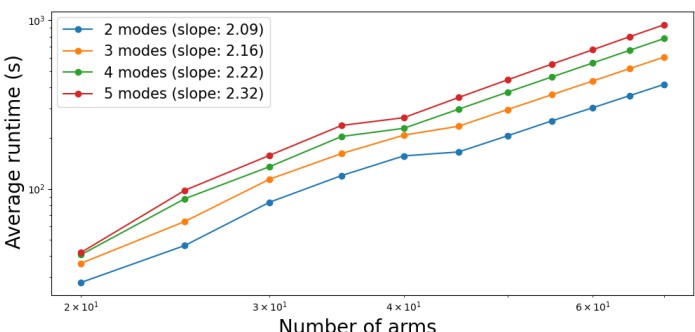

Figure 5: Runtime as a function of the number of arms.

All experiments were run on a single desktop PC equipped with an AMD Ryzen 7 5800X 8-core/16-thread CPU @ 3.8 GHz, 16 GB DDR4 RAM.

# B    Proofs of Section 3

*Proof of Proposition 1.* Applying Theorem 1 in [1] to the parameter set $\mathcal{F}_{\leq m}$ yields that the Graves-Lai constant $C(m, \boldsymbol{\mu})$ is the value of

$$\text{minimize}_{\boldsymbol{\eta}} \ \boldsymbol{\eta}^\top \boldsymbol{\Delta} \text{ subject to } \inf_{\boldsymbol{\lambda} \in \boldsymbol{\lambda}(m, \boldsymbol{\mu})} \boldsymbol{\eta}^\top d(\boldsymbol{\mu}, \boldsymbol{\lambda}) \geq 1 \text{ and } \boldsymbol{\eta} \geq 0$$

with $\boldsymbol{\lambda}(m, \boldsymbol{\mu}) = \{\boldsymbol{\lambda} \in \mathcal{F}_{\leq m}, d_k(\mu_k, \lambda_k) = 0, k^\star(\boldsymbol{\lambda}) \neq k\}$ for $k = k^\star(\boldsymbol{\mu})$. By Assumption 1, $d_k(\mu_k, \lambda_k) = 0$ holds if and only if $\mu_k = \lambda_k$, which concludes the proof. $\square$

# C    Proofs of Section 4

## C.1    Proof of Proposition 2

Assume that $B(m, \boldsymbol{\mu}) = \cup_{i=1}^{\mathcal{U}(K,m)} Z_i$ where the $Z_i$ are disjoint convex sets. Denote by $\mathcal{X}(m)$ the set of independent sets of $G$ of size $m$, and let us lower bound its size. Consider the process in which we first select a node $x_1 \in [K]$, and then for $i = 2, \ldots, m$ we select a node $x_i \in [K]$ in $[K] \setminus \cup_{j=1}^{i-1} \mathcal{N}(x_j)$ where $\mathcal{N}(x_j)$ is $x_j$ and its neighbors. Then $X = \{x_1, \ldots, x_m\}$ is an independent set of $G$ of size $m$, and at step $i$ of the process has at least

$$|[K] \setminus \cup_{j=1}^{i-1} N(x_j)| \geq K - \sum_{j=1}^{i-1} |N(x_j)| \geq K - m(\deg(G) + 1)$$

choices, since $|N(x_j)| \leq \deg(G) + 1$ and $i \leq m$. Hence, the number of independent sets of $G$ of size $m$ is at least

$$|\mathcal{X}(m)| \geq \frac{1}{m!}(K - (\deg(G) + 1)m)^m.$$

For each independent set of size $m$, $X \in \mathcal{X}(m)$, define the vector $\boldsymbol{\mu}^X \in \mathbb{R}^K$ such that $\mu_k^X = \mathbb{1}\{k \in X\}$ for $k \in [K]$. We can readily check that $\boldsymbol{\mu}^X$ is multimodal with $m$ modes, and that its $m$ modes are precisely the elements of $X$. Now consider another independent set of size $m$, $X' \in \mathcal{X}(m)$ and consider $\boldsymbol{\lambda}^{X,X'} = (3/4)\boldsymbol{\mu}^X + (1/4)\boldsymbol{\mu}^{X'}$ a convex combination of $\boldsymbol{\mu}^X$ and $\boldsymbol{\mu}^{X'}$. For $k \in X$ we have that $\lambda_k = 3/4$ and $\lambda_{k'} \leq 1/4$ for any $k'$ that is a neighbor of $k$, and hence $k$ is a mode of $\boldsymbol{\lambda}$. On the other hand, assume that there exists $k \in X'$ such that $k$ is not a neighbor of a node in $X$. Then $\lambda_k^{X,X'} = 1/4$ and $\lambda_{k'}^{X,X'} = 0$ for any $k'$ that is a neighbor of $k$ so that $k$ is a mode of $\boldsymbol{\lambda}^{X,X'}$. Putting it together, this means that, for $\boldsymbol{\lambda}^{X,X'}$ to have $m$ modes, one must make sure each element of $X'$ is the neighbor of an element in $X$.

Consider $X \in Z_i$ for some $i$. For any $X' \in \cup Z_i$ by convexity we must have $\boldsymbol{\lambda}^{X,X'} \in Z_i$ so that $\boldsymbol{\lambda}^{X,X'}$ has $m$ modes. By the above this means that all elements of $X'$ must be neighbors of $X$, so $|\mathcal{X}(m) \cup Z_i| \leq \binom{\deg(G)m}{m}$ since $X'$ has $m$ elements and $X$ has at most $\deg(G)m$ neighbors. Since the $Z_i$ are disjoint,

$$\frac{1}{m!}(K - (\deg(G) + 1)m)^m \leq |\mathcal{X}(m)| = \sum_{i=1}^{\mathcal{U}(K,m)} |\mathcal{X}(m) \cup Z_i| \leq \mathcal{U}(K,m)\binom{\deg(G)m}{m}.$$

This yields the result: $\mathcal{U}(K,m) \geq \frac{((\deg(G)-1)m)!}{(\deg(G)m)!}(K - (\deg(G) + 1)m)^m$.

## C.2  Proof of Proposition 3

By definition, for any $\boldsymbol{\lambda} \in \mathcal{B}_k(m, \boldsymbol{\mu})$, we must have $\lambda_k > \mu^\star$, so that $\inf_{\boldsymbol{\lambda} \in B(m,\boldsymbol{\mu})} \boldsymbol{\eta}^\top d(\boldsymbol{\mu}, \boldsymbol{\lambda}) \geq \eta_k d_k(\mu_k, \mu^\star)$. Conversely, let $\varepsilon > 0$ and $\boldsymbol{\lambda}^\varepsilon = \boldsymbol{\mu} + (\mu^\star + \varepsilon - \mu_k)\boldsymbol{e}^{(k)}$. If $|\mathcal{M}(\boldsymbol{\mu})| < m$, $\mathcal{M}(\boldsymbol{\lambda}^\varepsilon) \subset \mathcal{M}(\boldsymbol{\mu}) \cup \{k\}$ ensures that $|\mathcal{M}(\boldsymbol{\lambda}^\varepsilon)| \leq m$ and in turn $\boldsymbol{\lambda}^\varepsilon \in \mathcal{B}_k(m, \boldsymbol{\mu})$. If $k \in \mathcal{N}(\boldsymbol{\mu})$, either $k$ is a mode of $\boldsymbol{\mu}$ and $\mathcal{M}(\boldsymbol{\lambda}^\varepsilon) = \mathcal{M}(\boldsymbol{\mu})$, or $k$ is a neighbor of a mode $\ell$ and $\mathcal{M}(\boldsymbol{\lambda}^\varepsilon) = \mathcal{M}(\boldsymbol{\mu}) \cup \{k\} \setminus \{\ell\}$. In any case, $|\mathcal{M}(\boldsymbol{\lambda}^\varepsilon)| \leq m$ and in turn, $\boldsymbol{\lambda}^\varepsilon \in \mathcal{B}_k(m, \boldsymbol{\mu})$. Consequently, $\inf_{\boldsymbol{\lambda} \in \mathcal{B}_k(m,\boldsymbol{\mu})} \boldsymbol{\eta}^\top d(\boldsymbol{\mu}, \boldsymbol{\lambda}) \leq \boldsymbol{\eta}^\top d(\boldsymbol{\mu}, \boldsymbol{\lambda}^\varepsilon) = \eta_k d_k(\mu_k, \mu^\star + \varepsilon)$. Letting $\varepsilon \to 0$ yields the other side of the inequality. Finally, note that $\boldsymbol{\mu} + (\mu^\star - \mu_k)\boldsymbol{e}^{(k)} = \lim_{\varepsilon \to 0} \boldsymbol{\lambda}^\varepsilon \in \overline{\mathcal{B}_k}(m, \boldsymbol{\mu})$. This concludes the proof.

## C.3  Proof of Proposition 4

We first demonstrate the following intermediary result.

**Lemma 1.** *The closure of $\mathcal{B}_k(m, \boldsymbol{\mu})$ in $\mathbb{R}^K$ is given by*

$$F_k = \{\boldsymbol{\lambda} \in \mathcal{F}_{\leq m}, \lambda_{k^\star(\boldsymbol{\mu})} = \mu^\star, \lambda_k = \max_{j \in [K]} \lambda_j, \lambda_k \geq \mu^\star\}.$$

*Proof.* Consider $(\boldsymbol{\lambda}^t)_{t \geq 0}$ a sequence of elements of $\mathcal{B}_k(m, \boldsymbol{\mu})$ with $\lim_{t \to \infty} \boldsymbol{\lambda}^t = \boldsymbol{\lambda}^\infty$ and let us prove that $\boldsymbol{\lambda}^\infty \in F_k$. For all $t \geq 0$, we have $\lambda_{k^\star(\boldsymbol{\mu})}^t = \lambda_{k^\star(\boldsymbol{\mu})}^\infty = \mu^\star$. Furthermore, for all $t \geq 0$, $k^\star(\boldsymbol{\lambda}^t) = k$, hence $\lambda_k^t > \lambda_{k^\star(\boldsymbol{\mu})}^t = \mu^\star$ so that $\lambda_k^\infty \geq \mu^\star$. Now, let $i \in \mathcal{M}(\boldsymbol{\lambda}^\infty)$, which implies $\lambda_i^\infty > \lambda_\ell^\infty$ for all $\ell$ neighbor of $i$. Hence, for all $t$ large enough, $i \in \mathcal{M}(\boldsymbol{\lambda}^t)$. Repeating the same reasoning for all the modes of $\boldsymbol{\lambda}^\infty$, for all $t$ large enough, $\mathcal{M}(\boldsymbol{\lambda}^\infty) \subset \mathcal{M}(\boldsymbol{\lambda}^t)$, and $|\mathcal{M}(\boldsymbol{\lambda}^\infty)| \leq |\mathcal{M}(\boldsymbol{\lambda}^t)| \leq m$. We have proven that $\boldsymbol{\lambda}^\infty \in F_k$. Conversely, let $\boldsymbol{\lambda} \in F_k$. Either $\lambda_k > \mu^\star$ and $\boldsymbol{\lambda} \in \mathcal{B}_k(m, \boldsymbol{\mu}) \subset \overline{\mathcal{B}_k}(m, \boldsymbol{\mu})$, or $\lambda_j \leq \mu^\star$ for all $j$. There are then two cases to distinguish.

(i) If $k$ is a mode of $\boldsymbol{\lambda}$, we can write $\boldsymbol{\lambda} = \lim_{t \to \infty} \boldsymbol{\lambda}^t$ where $\boldsymbol{\lambda}^t \in \mathcal{B}_k(m, \boldsymbol{\mu})$ is defined by $\lambda_k^t = \mu^\star + 1/t$ and $\lambda_j^t = \lambda_j$ for $j \neq k$.

(ii) If $k$ is not a mode of $\boldsymbol{\lambda}$, as $\lambda_j \leq \mu^\star$ for all $j$, this must mean that there exists a neighbor $\ell$ of $k$ such that $\lambda_\ell = \lambda_k = \mu^\star$. Note further that $k \notin \mathcal{N}(\boldsymbol{\mu})$ ensures $\ell \neq k^\star(\boldsymbol{\mu})$. Then we can

write $\boldsymbol{\lambda} = \lim_{t\to\infty} \boldsymbol{\lambda}^t$ where $\boldsymbol{\lambda}^t \in \mathcal{B}_k(m, \boldsymbol{\mu})$ is defined by $\lambda_j^t = \mu^\star + 1/t$ for $j \in \{k, \ell\}$ and $\lambda_j^t = \lambda_j$ otherwise.

In any case, we have shown that $\boldsymbol{\lambda} \in \overline{B}_k(m, \boldsymbol{\mu})$, which concludes the proof. $\hfill\square$

We now prove Proposition 4. By Assumption 2, $\boldsymbol{\lambda} \mapsto \boldsymbol{\eta}^\top d(\boldsymbol{\mu}, \boldsymbol{\lambda})$ is continuous, so that $\inf_{\boldsymbol{\lambda}\in\mathcal{B}_k(m,\boldsymbol{\mu})} \boldsymbol{\eta}^\top d(\boldsymbol{\mu}, \boldsymbol{\lambda}) = \inf_{\boldsymbol{\lambda}\in\overline{\mathcal{B}_k}(m,\boldsymbol{\mu})} \boldsymbol{\eta}^\top d(\boldsymbol{\mu}, \boldsymbol{\lambda})$. Consider now $\boldsymbol{\lambda} \in \overline{\mathcal{B}_k}(m, \boldsymbol{\mu})$ such that $\lambda_\ell > \mu^\star$ for some $\ell$ and let $\varepsilon = \min_{\ell, \lambda_\ell > \mu^\star}(\lambda_\ell - \mu^\star) > 0$ the minimal amount by which an entry of $\boldsymbol{\lambda}$ is larger than $\mu^\star$. We will show that there exists $\boldsymbol{\lambda}' \in \overline{\mathcal{B}_k}(m, \boldsymbol{\mu})$ such that $\boldsymbol{\eta}^\top d(\boldsymbol{\mu}, \boldsymbol{\lambda}') < \boldsymbol{\eta}^\top d(\boldsymbol{\mu}, \boldsymbol{\lambda})$.

Consider a graph $G' = ([K], E')$ where $(i, j) \in E'$ if and only if $(i, j) \in E$ and $\lambda_i = \lambda_j$. Consider $S \subset [K]$ the connected component of $G'$ where $\ell$ lies. Consider the minimal gap between two neighboring arms: $\delta = \min_{(i,j)\in E, \lambda_i \neq \lambda_j} |\lambda_i - \lambda_j|$. Consider $\boldsymbol{\lambda}'$ such that $\lambda_i' = \lambda_i - (\min(\varepsilon, \delta)/2)\mathbb{1}\{i \in S\}$. As the nodes are modified by strictly less than $\delta$, we have $\mathcal{M}(\boldsymbol{\lambda}') = \mathcal{M}(\boldsymbol{\lambda})$. As they are modified by strictly less than $\varepsilon$, $\lambda_k' > \mu^\star$. In turn, $\boldsymbol{\lambda}' \in \overline{\mathcal{B}_k}(m, \boldsymbol{\mu})$. Furthermore, $d(\boldsymbol{\mu}, \boldsymbol{\lambda}') < d(\boldsymbol{\mu}, \boldsymbol{\lambda})$ since for all $k$, $\lambda_k \mapsto d_k(\mu_k, \lambda_k)$ is strictly increasing whenever $\lambda_k > \mu^\star \geq \mu_k$, which implies that $\boldsymbol{\eta}^\top d(\boldsymbol{\mu}, \boldsymbol{\lambda}') < \boldsymbol{\eta}^\top d(\boldsymbol{\mu}, \boldsymbol{\lambda})$.

We may prove similarly that if $\lambda_\ell < \mu_\star$ for some $\ell$, there exists $\boldsymbol{\lambda}' \in \overline{\mathcal{B}_k}(m, \boldsymbol{\mu})$ such that $\boldsymbol{\eta}^\top d(\boldsymbol{\mu}, \boldsymbol{\lambda}') < \boldsymbol{\eta}^\top d(\boldsymbol{\mu}, \boldsymbol{\lambda})$. This ensures that $\inf_{\boldsymbol{\lambda}\in\overline{\mathcal{B}_k}(m,\boldsymbol{\mu})} \boldsymbol{\eta}^\top d(\boldsymbol{\mu}, \boldsymbol{\lambda}) = \inf_{\boldsymbol{\lambda}\in\mathcal{B}_k'(m,\boldsymbol{\mu})} \boldsymbol{\eta}^\top d(\boldsymbol{\mu}, \boldsymbol{\lambda})$. Finally, as $\boldsymbol{\lambda} \mapsto \boldsymbol{\eta}^\top d(\boldsymbol{\mu}, \boldsymbol{\lambda})$ is continuous on the compact set $\mathcal{B}_k'(m, \boldsymbol{\mu})$, the infimum is attained. This concludes the proof.

## C.4 Proof of Proposition 5

Consider $\boldsymbol{\eta}$ defined as $\eta_k = \frac{1}{d_k(\mu_k,\mu^\star)}$ for all $k \neq k^\star(\boldsymbol{\mu})$, and where the value of $\eta_{k^\star(\boldsymbol{\mu})}$ is arbitrary. Let us check that $\boldsymbol{\eta}$ is a feasible solution to $P_{GL}$. For any $\boldsymbol{\lambda} \in B(m, \boldsymbol{\mu})$, there must exist $k \neq k^\star(\boldsymbol{\mu})$ such that $\lambda_k > \mu_\star$ and therefore $\boldsymbol{\eta}^\top d(\boldsymbol{\mu}, \boldsymbol{\lambda}) \geq \eta_k d_k(\mu_k, \lambda_k) > \eta_k d_k(\mu_k, \mu^\star) = 1$. This shows that $\inf_{\boldsymbol{\lambda}\in B(m,\boldsymbol{\mu})} \boldsymbol{\eta}^\top d(\boldsymbol{\mu}, \boldsymbol{\lambda}) \geq 1$ hence $\boldsymbol{\eta}$ is indeed a feasible solution to $P_{GL}$, so $\boldsymbol{\eta}$ must have a higher value than a solution $\boldsymbol{\eta}^\star$ of $P_{GL}$. As long as $\eta_k^\star = 0$ when $\Delta_k = 0$ (note that this choice does not impact the value of $P_{GL}$), we get

$$\|\boldsymbol{\eta}^\star\|_\infty \Delta_{\min} \leq \boldsymbol{\eta}^{\star\top}\boldsymbol{\Delta} \leq \boldsymbol{\eta}^\top\boldsymbol{\Delta} = \sum_{k, \Delta_k > 0} \frac{\Delta_k}{d_k(\mu_k, \mu^\star)}$$

hence $\|\boldsymbol{\eta}^\star\|_\infty \leq \mathfrak{B}(\boldsymbol{\mu})$ as announced.

## C.5 Proof of Proposition 6

Consider $i \neq k$ a mode $i \in \mathcal{M}(\boldsymbol{\lambda}^\star)$, and define the minimal difference between $i$ and its neighbors $\delta_i = \min_{(i,j)\in E} |\lambda_i^\star - \lambda_j^\star|)$ For all $\delta' \in (-\delta_i, \delta_i)$, it is noted that $\mathcal{M}(\boldsymbol{\lambda}^\star) = \mathcal{M}(\boldsymbol{\lambda}^\star + \delta' e^{(i)})$, and so $\boldsymbol{\lambda}^\star + \delta' e^{(i)} \in \mathcal{B}_k'(m, \boldsymbol{\mu})$. Therefore, the function $\delta' \mapsto \boldsymbol{\eta} d(\mu^\star, \boldsymbol{\lambda}^\star + \delta' e^{(i)})$ must attain its minimum at $\delta' = 0$, which implies $\lambda_i^\star = \mu_i$. Further consider $\ell$ a neighbor of $i$ and $\delta_\ell = |\lambda_i^\star - \lambda_\ell^\star|$. For all $\delta' \in [0, \delta_\ell)$, it is noted that $\mathcal{M}(\boldsymbol{\lambda}^\star + \delta' e^{(\ell)}) \subset \mathcal{M}(\boldsymbol{\lambda}^\star)$, so that $\boldsymbol{\lambda}^\star + \delta' e^{(\ell)} \in \mathcal{B}_k'(m, \boldsymbol{\mu})$. Therefore, the function $\delta' \mapsto \boldsymbol{\eta} d(\mu^\star, \boldsymbol{\lambda}^\star + \delta' e^{(\ell)})$ must attain its minimum at $\delta' = 0$, which implies $\lambda_\ell^\star \geq \mu_\ell$. Putting it together, we have proven that, if $i \in \mathcal{M}(\boldsymbol{\lambda}^\star)$, for all $(\ell, i) \in E$ we have $\mu_\ell \leq \lambda_\ell^\star < \lambda_i^\star = \mu_i$, and in turn $i \in \mathcal{M}(\boldsymbol{\mu})$, which concludes the proof.

## C.6 Proof of Proposition 7

Let us consider $\boldsymbol{\lambda}$ a minimizer of $\boldsymbol{\eta}^\top d(\boldsymbol{\mu}, \boldsymbol{\lambda})$ over $\mathcal{B}_{k,k'}(m, \boldsymbol{\mu})$. We already know that $\boldsymbol{\lambda} \in [\mu_\star, \mu^\star]$ from Proposition 4. Consider $G^k$ the directed tree rooted at $k$ and define $\tilde{\boldsymbol{\lambda}}$ a rounding of $\boldsymbol{\lambda}$ using the following recursive procedure: start at the root $\tilde{\lambda}_k \in \arg\min\{|x - \lambda_k| : x \in D(n, \boldsymbol{\mu})\}$ then for all $\ell$, choose

$$\tilde{\lambda}_\ell = \arg\min\{|x - \tilde{\lambda}_{p(\ell)}| : x \in D(n, \boldsymbol{\mu}), \text{sgn}(x - \tilde{\lambda}_{p(\ell)}) = \text{sgn}(\lambda_\ell - \lambda_{p(\ell)})\}$$

where $p(\ell)$ is the parent of $\ell$ and $\mathrm{sgn}$ is the sign function. The idea is that, when rounding in this fashion, we both have the insurance that for any $(i, \ell) \in E$ $|\tilde{\lambda}_\ell - \tilde{\lambda}_i| \leq (1/n)(\mu^\star - \mu_\star)$ so that, by recursion over $\ell$: $\|\tilde{\boldsymbol{\lambda}} - \boldsymbol{\lambda}\|_\infty \leq (1/n)\mathrm{diam}(G)(\mu^\star - \mu_\star)$ and we also have that $\mathrm{sgn}(\lambda_i - \lambda_\ell) = \mathrm{sgn}(\tilde{\lambda}_i - \tilde{\lambda}_\ell)$ so that $M(\tilde{\boldsymbol{\lambda}}) = \mathcal{M}(\boldsymbol{\lambda})$. In essence, we round $\boldsymbol{\lambda}$ to ensure both a small rounding error while leaving the set of modes unchanged. We further have

$$
\begin{aligned}
\boldsymbol{\eta}^\top d(\boldsymbol{\mu}, \tilde{\boldsymbol{\lambda}}) &= \boldsymbol{\eta}^\top d(\boldsymbol{\mu}, \tilde{\boldsymbol{\lambda}}) + \boldsymbol{\eta}^\top (d(\boldsymbol{\mu}, \tilde{\boldsymbol{\lambda}}) - d(\boldsymbol{\mu}, \boldsymbol{\lambda})) \leq \boldsymbol{\eta}^\top d(\boldsymbol{\mu}, \tilde{\boldsymbol{\lambda}}) + \|\boldsymbol{\eta}\|_\infty \| d(\boldsymbol{\mu}, \tilde{\boldsymbol{\lambda}}) - d(\boldsymbol{\mu}, \boldsymbol{\lambda})\|_1 \\
&\leq \boldsymbol{\eta}^\top d(\boldsymbol{\mu}, \tilde{\boldsymbol{\lambda}}) + \mathfrak{B}(\boldsymbol{\mu})\mathfrak{A}(\boldsymbol{\mu})\|\tilde{\boldsymbol{\lambda}} - \boldsymbol{\lambda}\|_1 \leq \boldsymbol{\eta}^\top d(\boldsymbol{\mu}, \tilde{\boldsymbol{\lambda}}) + \mathfrak{B}(\boldsymbol{\mu})\mathfrak{A}(\boldsymbol{\mu})K\|\tilde{\boldsymbol{\lambda}} - \boldsymbol{\lambda}\|_\infty \\
&\leq \boldsymbol{\eta}^\top d(\boldsymbol{\mu}, \tilde{\boldsymbol{\lambda}}) + \frac{\mathfrak{C}(\boldsymbol{\mu})}{n}
\end{aligned}
$$

with $\mathfrak{C}(\boldsymbol{\mu}) = \mathrm{diam}(G)(\mu^\star - \mu_\star)\mathfrak{B}(\boldsymbol{\mu})\mathfrak{A}(\boldsymbol{\mu})K$ using Assumption 2, the fact that $\|\boldsymbol{\eta}\|_\infty \leq \mathfrak{B}(\boldsymbol{\mu})$ and the previous bound. This concludes the proof.

### C.7 Proof of Proposition 8

We recall that we consider the graph $G^k$, which is a directed tree rooted at $k$. Consider node $\ell$ and its parent $p(\ell)$, assume that $\lambda_{p(\ell)}$ is known, and we wish to minimize the value:

$$
\eta_\ell d_\ell(\mu_\ell, \lambda_\ell) + \sum_{j \in \mathcal{D}(\ell)} \eta_j d_j(\mu_j, \lambda_j)
$$

which corresponds to $\boldsymbol{\eta}^\top d(\boldsymbol{\mu}, \boldsymbol{\lambda})$ restricted to $\ell$ and its descendants, subject to $\boldsymbol{\lambda} \in \mathcal{B}_{k,k'}(m, \boldsymbol{\mu})$. Then it suffices to optimize over $\lambda_\ell$ and $\lambda_j$ for $j \in \mathcal{D}(\ell)$. Also, we may readily check that setting $\eta_{k^\star(\boldsymbol{\mu})} = +\infty$ is equivalent to enforcing the constraint $\lambda_{k^\star(\boldsymbol{\mu})} = \mu^\star$. Of course, the sign of $\lambda_\ell - \lambda_{p(\ell)}$ is important to ensure that the constraints are satisfied. Define $f_\ell(z, +1)$ the minimal value that can be obtained if selecting $\lambda_\ell = z > \lambda_{p(\ell)}$ and $f_\ell(z, -1)$ the minimal value that can be obtained if selecting $\lambda_\ell = z \leq \lambda_{p(\ell)}$. Consider $\lambda_\ell = z$ and $\lambda_{p(\ell)}$ fixed, and let us examine how one can select $\lambda_j$ for $j \in \mathcal{C}(\ell)$ the children of $\ell$ to ensure that $\boldsymbol{\lambda} \in \mathcal{B}_{k,k'}(m, \boldsymbol{\mu})$ is respected.

(i) First consider $\ell \notin \mathcal{M}(\boldsymbol{\mu}) \cup \{k\} \setminus \{k'\}$, so that $\ell$ should not be a mode. If $\lambda_\ell \leq \lambda_{p(\ell)}$ then $\ell$ cannot be a mode anyways, so for each $j \in \mathcal{C}(\ell)$ we have two choices: select $\lambda_j \leq \lambda_\ell$, which gives a minimal value of $\min_{w \leq z} f_j(w, -1) = f_j^\star(z, -1)$, or select $\lambda_j > \lambda_\ell$, which gives a minimal value of $\min_{w > z} f_j(w, +1) = f_j^\star(z, +1)$. The minimal value over these two choices is $f_j^\diamond(z) = \min(f_j^\star(z, -1), f_j^\star(z, +1))$. Therefore,

$$
f_\ell(z, -1) = \eta_\ell d_\ell(\mu_\ell, z) + \sum_{j \in \mathcal{C}(\ell)} f_j^\diamond(z).
$$

(ii) If $\lambda_\ell > \lambda_{p(\ell)}$, since $\ell$ should not be a mode, one must make sure that there exists at least one child $v \in \mathcal{C}(\ell)$ of $\ell$ such that $\lambda_\ell \leq \lambda_v$, and the value of $\lambda_j$ for $j \in \mathcal{C}(\ell) \setminus \{v\}$ can be chosen freely. Of course, if $\mathcal{C}(\ell) = \emptyset$ this is not possible and one has simply $f_\ell(z, +1) = +\infty$. If $\mathcal{C}(\ell) \neq \emptyset$ and $v \in \mathcal{C}(\ell)$, $\min\{f_v^\star(z, +1), f_v(z, -1)\}$ represents the minimal cost of choosing a value of $\lambda_v$ such that $\lambda_\ell \leq \lambda_v$. By the same reasoning as before, the minimal obtainable value is therefore:

$$
f_\ell(z, +1) = \eta_\ell d_\ell(\mu_\ell, z) + \sum_{j \in \mathcal{C}(\ell)} f_j^\diamond(z) + \min_{v \in \mathcal{C}(\ell)} g_v(z)
$$

for $g_v(z) = \min\{f_v^\star(z, +1), f_v(z, -1)\} - f_v^\diamond(z)$, where we have minimized over the choice of $v \in \mathcal{C}(\ell)$.

(iii) Now consider the case $\ell \in \mathcal{M}(\boldsymbol{\mu}) \cup \{k\} \setminus \{k'\}$. Since $\ell$ can either be a mode or not a mode, regardless of the sign of $\lambda_\ell - \lambda_{p(\ell)}$, we have no constraints on the choice of $\lambda_j$ for $j \in \mathcal{C}(\ell)$ and the minimal value that can be obtained is

$$
f_\ell(z, u) = \eta_\ell d_\ell(\mu_\ell, z) + \sum_{j \in \mathcal{C}(\ell)} f_j^\diamond(z)
$$

for both $u = -1$ and $u = +1$.

We have proven that $f_\ell$ indeed obeys the dynamic programming equations. Since $k$ is the root of the tree and we must have $\lambda_k^\star = \mu^\star$, then $f_k(\mu^\star, u)$ is the value of $P_{GL}(k, k')$ for any $u$.

## C.8 Proof of Corollary 1

From the definition of $\tilde{P}_{GL}(k, k')$, $\lambda_k^\star = \mu^\star$. Consider $\ell \neq k$ and its parent $p(\ell)$ in $G^k$, and assume that $\lambda_{p(\ell)}^\star$ is known. There are two cases to consider:

(i) If $p(\ell) \notin \mathcal{M}(\boldsymbol{\mu}) \cup \{k\} \setminus \{k'\}$, $\lambda_{p(\ell)}^\star > \lambda_{p^2(\ell)}^\star$ where $p^2(\ell)$ is the parent of $p(\ell)$, and $\ell = \arg\min_{v \in \mathcal{C}(p(\ell))} g_v(z)$ is the node that induces the smallest cost when constrained, we must have $\lambda_\ell^\star \geq \lambda_p^\star(\ell)$ to ensure that $p(\ell)$ is not a mode. There are two choices: either select $\lambda_\ell^\star > \lambda_{p(\ell)}^\star$ which yields value $f_\ell^\star(\lambda_{p(\ell)}^\star, +1)$ and dictates the choice $\lambda_\ell^\star \in \arg\min_{z > \lambda_{p(\ell)}^\star} f_\ell(z, +1)$, otherwise select $\lambda_\ell^\star = \lambda_{p(\ell)}^\star$ which yields value $f_\ell(\lambda_{p(\ell)}^\star, -1)$. Taking the minimal value among the two choices gives:

$$\lambda_\ell^\star = \begin{cases} \arg\min_{z > \lambda_{p(\ell)}^\star} f_\ell(z, +1) & \text{if } f_\ell^\star(\lambda_{p(\ell)}^\star, +1) \leq f_\ell(\lambda_{p(\ell)}^\star, -1) \\ \lambda_{p(\ell)}^\star & \text{if } f_\ell^\star(\lambda_{p(\ell)}^\star, +1) > f_\ell(\lambda_{p(\ell)}^\star, -1) \end{cases}$$

(ii) Otherwise, there are no constraints on the value of $\lambda_\ell^\star$: either select $\lambda_\ell^\star > \lambda_{p(\ell)}^\star$ which yields value $f_\ell^\star(\lambda_{p(\ell)}^\star, +1)$ and dictates the choice $\lambda_\ell^\star \in \arg\min_{z > \lambda_{p(\ell)}^\star} f_\ell(z, +1)$, otherwise select $\lambda_\ell^\star \leq \lambda_{p(\ell)}^\star$ which yields value $f_\ell^\star(\lambda_{p(\ell)}^\star, -1)$ and dictates the choice $\lambda_\ell^\star \in \arg\min_{z \leq \lambda_{p(\ell)}^\star} f_\ell(z, -1)$. Taking the minimal value among the two choices gives:

$$\lambda_\ell^\star = \begin{cases} \arg\min_{z > \lambda_{p(\ell)}^\star} f_\ell(z, +1) & \text{if } f_\ell^\star(\lambda_{p(\ell)}^\star, +1) \leq f_\ell^\star(\lambda_{p(\ell)}^\star, -1) \\ \arg\min_{z \leq \lambda_{p(\ell)}^\star} f_\ell(z, -1) & \text{if } f_\ell^\star(\lambda_{p(\ell)}^\star, +1) > f_\ell^\star(\lambda_{p(\ell)}^\star, -1) \end{cases}$$

It is noted that storing the values of $f_\ell(z, u)$, $f_\ell^\star(z, u)$ and $f_\ell^\diamond(z)$ for $\ell \in [K]$, $z \in D(n, \boldsymbol{\mu})$ and $u \in \{-1, +1\}$ requires memory $O(nK)$ since $|D(n, \boldsymbol{\mu})| = n$. Furthermore, if $f_j(z, u)$, $f_j^\star(z, u)$ and $f_j^\diamond(z)$ for $j \in \mathcal{C}(\ell)$, $z \in D(n, \boldsymbol{\mu})$ and $u \in \{-1, +1\}$, have been computed, then one may compute $f_\ell(z, u)$, $f_\ell^\star(z, u)$ and $f_\ell^\diamond(z)$ for $z \in D(n, \boldsymbol{\mu})$ and $u \in \{-1, +1\}$ in time $O(n|\mathcal{C}(\ell)|)$ from the dynamic programming equations. Since $\sum_{\ell \in [K]} |\mathcal{C}(\ell)| = K - 1$, the complete procedure requires time $O(nK)$. This completes the proof.

## C.9 Proof of Proposition 9

To ease the notation, denote

$$g(\boldsymbol{\eta}, \boldsymbol{\mu}, n) = \min_{\boldsymbol{\lambda} \in \overline{B}(m, \boldsymbol{\mu}) \cap D(n, \boldsymbol{\mu})^K} \{\boldsymbol{\eta}^\top d(\boldsymbol{\mu}, \boldsymbol{\lambda})\} \text{ and } g(\boldsymbol{\eta}, \boldsymbol{\mu}) = \min_{\boldsymbol{\lambda} \in \overline{B}(m, \boldsymbol{\mu})} \{\boldsymbol{\eta}^\top d(\boldsymbol{\mu}, \boldsymbol{\lambda})\}.$$

Recall that $h(\boldsymbol{\eta}) = \boldsymbol{\eta}^\top \boldsymbol{\Delta} + \gamma \max[1 - g(\boldsymbol{\eta}, \boldsymbol{\mu}, n), 0]$ which is convex as a sum of a linear function, and a maximum of linear functions, with subgradients:

$$\boldsymbol{\Delta} - \gamma d(\boldsymbol{\mu}, \boldsymbol{\lambda}) \mathbb{1}\{g(\boldsymbol{\eta}, \boldsymbol{\mu}, n) < 1\} = v \in \partial h(\boldsymbol{\eta})$$

for any $\boldsymbol{\lambda}$ such that $g(\boldsymbol{\eta}, \boldsymbol{\mu}, n) = \boldsymbol{\eta}^\top d(\boldsymbol{\mu}, \boldsymbol{\lambda})$. The norm of a subgradient is upper bounded as

$$\|v\|_2 \leq \|\boldsymbol{\Delta}\|_2 + \gamma \|d(\boldsymbol{\mu}, \boldsymbol{\lambda})\|_2 \leq \|\boldsymbol{\Delta}\|_2 + \gamma K^{3/2} \mathfrak{A}(\boldsymbol{\mu})(\mu^\star - \mu_\star) = \mathfrak{E}(\boldsymbol{\mu})$$

since

$$\|d(\boldsymbol{\mu}, \boldsymbol{\lambda})\|_2 \leq \sqrt{K} \|d(\boldsymbol{\mu}, \boldsymbol{\lambda})\|_1 \leq \sqrt{K} \mathfrak{A}(\boldsymbol{\mu}) \|\boldsymbol{\mu} - \boldsymbol{\lambda}\|_1 \leq K^{3/2} \mathfrak{A}(\boldsymbol{\mu})(\mu^\star - \mu_\star).$$

Hence, $h$ is Lipschitz continuous with Lipschitz constant $\mathfrak{E}(\boldsymbol{\mu})$. Let $\boldsymbol{\eta}^\star$ the minimizer of $h$ over $(\mathbb{R}^+)^K$, and let us prove that $g(\boldsymbol{\eta}^\star, \boldsymbol{\mu}, n) \geq 1$. Any subgradient at $\boldsymbol{\eta}^\star$ must have positive entries so that:

$$0 \leq \boldsymbol{\Delta} - \gamma d(\boldsymbol{\mu}, \tilde{\boldsymbol{\lambda}}) \mathbb{1}\{\boldsymbol{\eta}^{\star\top} d(\boldsymbol{\mu}, \tilde{\boldsymbol{\lambda}}) \leq 1\}$$

where $\tilde{\boldsymbol{\lambda}}$ is such that $g(\boldsymbol{\eta}^\star, \boldsymbol{\mu}, n) = \boldsymbol{\eta}^{\star\top} d(\boldsymbol{\mu}, \tilde{\boldsymbol{\lambda}})$. In turn, since $\tilde{\boldsymbol{\lambda}} \in \overline{B}(m, \boldsymbol{\mu})$ there must exist $k \neq k^\star(\boldsymbol{\mu})$ such that $\tilde{\lambda}_k \geq \mu^\star$ and hence:

$$\mathbb{1}\{\boldsymbol{\eta}^\top d(\boldsymbol{\mu}, \tilde{\boldsymbol{\lambda}}) \leq 1\} \leq \frac{\Delta_k}{\gamma d_k(\mu_k, \mu^\star)} \leq \frac{1}{2}$$

by definition of $\gamma$. This implies that $\boldsymbol{\eta}^\top d(\boldsymbol{\mu}, \tilde{\boldsymbol{\lambda}}) \geq 1$ and so $g(\boldsymbol{\eta}^\star, \boldsymbol{\mu}, n) \geq 1$. So $\boldsymbol{\eta}^\star$ minimizes $\boldsymbol{\eta}^\top \boldsymbol{\Delta}$ over the set of $\boldsymbol{\eta}$ such that $g(\boldsymbol{\eta}, \boldsymbol{\mu}, n) \geq 1$.

We may now analyze the iterative scheme in itself. Since $h$ is convex and Lipschitz continuous with Lipschitz constant $\mathfrak{E}(\boldsymbol{\mu})$, and our iterative scheme is a projected subgradient descent with step size $\delta$, from [20][Lemma 14.1]:

$$h(\bar{\boldsymbol{\eta}}(t)) - h(\boldsymbol{\eta}^\star) \leq \frac{1}{2t}\left(\frac{\|\boldsymbol{\eta}^\star\|_2^2}{\delta} + t\delta\mathfrak{E}(\boldsymbol{\mu})^2\right) \leq \frac{1}{2t}\left(\frac{K\mathfrak{B}(\boldsymbol{\mu})^2}{\delta} + t\delta\mathfrak{E}(\boldsymbol{\mu})^2\right)$$

$$= \frac{\sqrt{K}\mathfrak{B}(\boldsymbol{\mu})\mathfrak{E}(\boldsymbol{\mu})}{\sqrt{t}} = \frac{\mathfrak{F}(\boldsymbol{\mu})}{\sqrt{t}}$$

where we used Proposition 5 and setting $\delta^2 = \frac{K\mathfrak{B}(\boldsymbol{\mu})^2}{t\mathfrak{E}(\boldsymbol{\mu})^2}$ to optimize the right hand side.

Let us now check that $\tilde{\boldsymbol{\eta}}(t)$ is an approximate solution to $P_{GL}$. From the above

$$\bar{\boldsymbol{\eta}}(t)^\top \boldsymbol{\Delta} \leq h(\bar{\boldsymbol{\eta}}(t)) \leq h(\boldsymbol{\eta}^\star) + \frac{\mathfrak{F}(\boldsymbol{\mu})}{\sqrt{t}} = \boldsymbol{\eta}^{\star\top}\boldsymbol{\Delta} + \frac{\mathfrak{F}(\boldsymbol{\mu})}{\sqrt{t}} \leq C(m, \boldsymbol{\mu}) + \frac{\mathfrak{F}(\boldsymbol{\mu})}{\sqrt{t}}$$

using the definition of $h$, the above bound, the fact that $\boldsymbol{\eta}^{\star\top}\boldsymbol{\Delta}$ is the minimum of $\boldsymbol{\eta}^\top\boldsymbol{\Delta}$ subject to $g(\boldsymbol{\eta}, \boldsymbol{\mu}, n) \geq 1$, $\boldsymbol{\eta} \geq 0$ and the fact that $C(m, \boldsymbol{\mu})$ is the minimum of $\boldsymbol{\eta}^\top\boldsymbol{\Delta}$ subject to $g(\boldsymbol{\eta}, \boldsymbol{\mu}) \geq 1$, $\boldsymbol{\eta} \geq 0$. Scaling the above on both sides gives a bound on the value of $\tilde{\boldsymbol{\eta}}(t)$:

$$\tilde{\boldsymbol{\eta}}(t)^\top \boldsymbol{\Delta} \leq \left(1 - \frac{\mathfrak{C}(\boldsymbol{\mu})}{n} - 2\frac{\mathfrak{F}(\boldsymbol{\mu})}{\gamma\sqrt{t}}\right)^{-1}\left(C(m, \boldsymbol{\mu}) + \frac{\mathfrak{F}(\boldsymbol{\mu})}{\sqrt{t}}\right)$$

We now have to check that $\tilde{\boldsymbol{\eta}}(t)$ is a feasible solution to $P_{GL}$. Denote by $\phi = g(\bar{\boldsymbol{\eta}}(t), \boldsymbol{\mu}, n)$. Consider the case $\phi < 1$, so that:

$$\bar{\boldsymbol{\eta}}(t)^\top \boldsymbol{\Delta} \geq \min_{\boldsymbol{\eta}:g(\boldsymbol{\eta}, \boldsymbol{\mu}, n) \geq \phi} \boldsymbol{\eta}^\top \boldsymbol{\Delta} = \phi \min_{\boldsymbol{\eta}:g(\boldsymbol{\eta}, \boldsymbol{\mu}, n) \geq 1} \boldsymbol{\eta}^\top \boldsymbol{\Delta} = \phi\boldsymbol{\eta}^{\star\top}\boldsymbol{\Delta}$$

where we used the fact that $g$ is homogeneous, i.e. $g(\boldsymbol{\eta}, \boldsymbol{\mu}, n) \geq \phi$ if and only if $g(\boldsymbol{\eta}/\phi, \boldsymbol{\mu}, n) \geq 1$. This implies

$$\phi\boldsymbol{\eta}^{\star\top}\boldsymbol{\Delta} + \gamma(1 - \phi) \leq \bar{\boldsymbol{\eta}}(t)^\top\boldsymbol{\Delta} + \gamma(1 - \phi) = h(\bar{\boldsymbol{\eta}}(t)) \leq h(\boldsymbol{\eta}^\star) + \frac{\mathfrak{F}(\boldsymbol{\mu})}{\sqrt{t}} = \boldsymbol{\eta}^{\star\top}\boldsymbol{\Delta} + \frac{\mathfrak{F}(\boldsymbol{\mu})}{\sqrt{t}}$$

Rearranging the terms we get:

$$1 - \phi \leq 2\frac{\mathfrak{F}(\boldsymbol{\mu})}{\gamma\sqrt{t}}$$

Therefore, using Proposition 7:

$$g(\bar{\boldsymbol{\eta}}(t), \boldsymbol{\mu}) \geq g(\bar{\boldsymbol{\eta}}(t), \boldsymbol{\mu}, n) - \frac{\mathfrak{C}(\boldsymbol{\mu})}{n} \geq 1 - \frac{\mathfrak{C}(\boldsymbol{\mu})}{n} - 2\frac{\mathfrak{F}(\boldsymbol{\mu})}{\gamma\sqrt{t}}$$

And once again using the homogeneity of $g$ we get

$$g(\tilde{\boldsymbol{\eta}}(t), \boldsymbol{\mu}) = g(\tilde{\boldsymbol{\eta}}(t), \boldsymbol{\mu}, n)\left(1 - \frac{\mathfrak{C}(\boldsymbol{\mu})}{n} - 2\frac{\mathfrak{F}(\boldsymbol{\mu})}{\gamma\sqrt{t}}\right) \geq 1$$

which is the announced result and concludes the proof.

# D  Proofs of Section 5

## D.1  Proof of Theorem 3

Consider $j \notin \mathcal{N}(\boldsymbol{\mu})$, $k \neq k^\star(\boldsymbol{\mu})$ a mode of $\boldsymbol{\mu}$ and $\ell$ the neighbor of $k$ which is the closest to $k$, that is $\arg\min_{\ell:(k,\ell)\in E}|\mu_k - \mu_\ell|$. Define the parameter $\boldsymbol{\lambda} = \boldsymbol{\mu} + (\mu^\star - \mu_j)e^{(j)} + (\mu_\ell - \mu_k)e^{(k)}$. One may check that $\boldsymbol{\lambda} \in \overline{B}(m, \boldsymbol{\mu})$. For any feasible local search strategy $\boldsymbol{\eta}$ we have $1 \leq \boldsymbol{\eta}^\top d(\boldsymbol{\mu}, \boldsymbol{\lambda}) = \eta_j d_j(\mu_j, \lambda_j) + \eta_k d_k(\mu_k, \lambda_\ell) = \eta_j d_j(\mu_j, \mu^\star) + \eta_k d_k(\mu_k, \mu_k - \delta_k) = \eta_k d_k(\mu_k, \mu_k - \delta_k)$ since $\eta_j = 0$

as $j \notin \mathcal{N}(\boldsymbol{\mu})$ and $\boldsymbol{\eta}$ is a local search strategy. Hence, $\Delta_k \eta_k \geq \frac{\Delta_k}{d_k(\mu_k, \mu_k - \delta_k)}$ for any mode $k$ and summing over modes we get the announced result

$$C_{\text{loc}}(m, \boldsymbol{\mu}) \geq \sum_{k \in \mathcal{M}(\boldsymbol{\mu}) \setminus \{k^\star(\boldsymbol{\mu})\}} \frac{\Delta_k}{d_k(\mu_k, \mu_k - \delta_k)}.$$

Now, since a multimodal bandit problem is also a classical bandit problem, we always have

$$C(m, \boldsymbol{\mu}) \leq \sum_{k \in [K]} \frac{\Delta_k}{d_k(\mu_k, \mu^\star)}$$

so that

$$\sup_{\boldsymbol{\mu} \in \mathcal{F}_m} \frac{C_{\text{loc}}(m, \boldsymbol{\mu})}{C(m, \boldsymbol{\mu})} = +\infty$$

as there exists reward functions $\boldsymbol{\mu} \in \mathcal{F}_m$ where $\delta_k$ is arbitrarily small while $\mu^\star - \mu_k$ remains comparatively large for any $k \neq k^\star(\boldsymbol{\mu})$, for instance consider a case where $\mu_k = 1$ if $k = k^\star(\boldsymbol{\mu})$, $\mu_k = \epsilon$ if $k \in \mathcal{M}(\boldsymbol{\mu}) \setminus \{k^\star(\boldsymbol{\mu})\}$ and $\mu_k = 0$ if $k \notin \mathcal{M}(\boldsymbol{\mu})$. This function has exactly $m$ modes and $\delta_k = \epsilon$ for any mode $k \neq k^\star(\boldsymbol{\mu})$.

## D.2 Suboptimality of `IMED-MB`

We argue that `IMED-MB` cannot be asymptotically optimal for all instances $\boldsymbol{\mu} \in \mathcal{F}_{\leq m}$, contrary to the statement of Theorem 2 in [18]. Consider an instance $\boldsymbol{\mu} \in \mathcal{F}_m$ for which the optimal value for local search strategies is strictly greater than the value of $P_{GL}$, i.e. $C_{\text{loc}}(m, \boldsymbol{\mu}) > C(m, \boldsymbol{\mu})$. Such an instance is guaranteed to exist by Theorem 3. Assume by contradiction that their Theorem 2 holds. Then, by their Proposition 1, `IMED-MB` is asymptotically optimal, so that

$$\limsup_{T \to \infty} \frac{R(\boldsymbol{\mu}, T)}{\ln(T)} \leq C(m, \boldsymbol{\mu}).$$

Furthermore, by their Proposition 1 and Theorem 2 again, for $k \neq k^\star(\boldsymbol{\mu})$ and $\eta_k = \lim_{T \to \infty} \frac{\mathbb{E}[N_k(T)]}{\ln(T)}$, we have $\eta_k = \frac{1}{d_k(\mu_k, \mu^\star)}$ if $k \in \mathcal{N}(\boldsymbol{\mu})$, and $\eta_k = 0$ otherwise. In particular, $\boldsymbol{\eta}$ is a local search strategy (and `IMED-MB` is uniformly good), so that its regret must verify

$$\liminf_{T \to \infty} \frac{R(\boldsymbol{\mu}, T)}{\ln T} \geq C_{\text{loc}}(m, \boldsymbol{\mu}).$$

Combining these inequalities leads to:

$$C_{\text{loc}}(m, \boldsymbol{\mu}) \leq \liminf_{T \to \infty} \frac{R(\boldsymbol{\mu}, T)}{\ln(T)} \leq \limsup_{T \to \infty} \frac{R(\boldsymbol{\mu}, T)}{\ln(T)} \leq C(m, \boldsymbol{\mu}),$$

a contradiction.

## D.3 Proof of Proposition 10

Consider $k$ a mode and $\ell$ one of its neighbors. For Gaussian rewards the conditions become $(\mu^\star - \mu_k)^2 \leq \kappa(\delta_k/2)^2$ and $(\mu^\star - \mu_\ell)^2 \leq \kappa(\mu_k - \delta_k/2 - \mu_\ell)^2$. These conditions are equivalent to $\Delta_k \leq \sqrt{\kappa}\delta_k/2$ and $\Delta_\ell \leq \sqrt{\kappa}(\Delta_\ell - \Delta_k - \delta_k/2)$. This must be true for all $\ell$ neighbor of $k$ and should hold when $\Delta_\ell = \Delta_k + \delta_k$, therefore we must have $\Delta_k \leq (\frac{\sqrt{\kappa}}{2} - 1)\delta_k$. Hence, $\boldsymbol{\mu}$ is $\kappa$-peaked if and only if $\Delta_k \leq \delta_k(\frac{\sqrt{\kappa}}{2} - 1)$ for all $k \in \mathcal{M}(\boldsymbol{\mu})$.

## D.4 Proof of Proposition 11

By Proposition 3, for $k \in \mathcal{N}(\boldsymbol{\mu}) \setminus \{k^\star(\boldsymbol{\mu})\}$, $\inf_{\boldsymbol{\lambda} \in B(m, \boldsymbol{\mu})} \boldsymbol{\eta}^\top d(\boldsymbol{\mu}, \boldsymbol{\lambda}) \leq \eta_k d_k(\mu_k, \mu^\star)$, so for any $\boldsymbol{\eta}$ feasible solution to $P_{GL}$ we must have $\eta_k \geq \frac{1}{d_k(\mu_k, \mu^\star)}$. Summing over $k \in \mathcal{N}(\boldsymbol{\mu}) \setminus \{k^\star(\boldsymbol{\mu})\}$, the value of $P_{GL}$ is lower bounded as

$$C(m, \boldsymbol{\mu}) \geq \sum_{k \in \mathcal{N}(\boldsymbol{\mu}) \setminus \{k^\star(\boldsymbol{\mu})\}} \frac{\Delta_k}{d_k(\mu_k, \mu^\star)}.$$

Now, consider the local search strategy $\boldsymbol{\eta}$ defined as

$$\eta_k = \begin{cases} \max\left(\frac{1}{d_k(\mu_k,\mu^\star)}, \frac{1}{d_k(\mu_k,\mu_k-\delta_k/2)}\right) & \text{if } k \in \mathcal{M}(\boldsymbol{\mu}) \setminus \{k^\star(\boldsymbol{\mu})\} \\ \max\left(\frac{1}{d_k(\mu_k,\mu^\star)}, \frac{1}{d_k(\mu_k,\mu_\ell-\delta_\ell/2)}\right) & \text{if } (k,\ell) \in E \text{ and } \ell \in \mathcal{M}(\boldsymbol{\mu}) \\ 0 & \text{otherwise.} \end{cases}$$

Let us check that $\boldsymbol{\eta}$ is feasible. Consider $\boldsymbol{\lambda} \in B(m,\boldsymbol{\mu})$ attaining its maximum at $k \in [K] \setminus \{k^\star(\boldsymbol{\mu})\}$. On the one hand, if $k \in \mathcal{N}(\boldsymbol{\mu})$, since $\lambda_k > \mu^\star$ we have $\boldsymbol{\eta}^\top d(\boldsymbol{\mu}, \boldsymbol{\lambda}) \geq \eta_k d_k(\mu_k, \lambda_k) > \eta_k d_k(\mu_k, \mu^\star) \geq 1$. On the other hand, if $k \notin \mathcal{N}(\boldsymbol{\mu})$ there must exist at least one $j \in \mathcal{M}(\boldsymbol{\mu})$ such that $j \notin \mathcal{M}(\boldsymbol{\lambda})$, as otherwise $\boldsymbol{\lambda}$ would have more than $m$ modes. In turn there must exist $\ell$ a neighbor of $j$ such that $\lambda_j \leq \lambda_\ell$. This implies that either $\lambda_j \leq \mu_j - \delta_j/2 < \mu_j$ and in that case $\boldsymbol{\eta}^\top d(\boldsymbol{\mu}, \boldsymbol{\lambda}) \geq \eta_j d_j(\mu_j, \lambda_j) \geq \eta_j d_j(\mu_j, \mu_j - \delta_j/2) \geq 1$, or $\lambda_\ell > \mu_j - \delta_j/2 > \mu_\ell$ and in that case $\boldsymbol{\eta}^\top d(\boldsymbol{\mu}, \boldsymbol{\lambda}) \geq \eta_\ell d_\ell(\mu_\ell, \lambda_\ell) > \eta_\ell d_\ell(\mu_\ell, \mu_j - \delta_j/2) \geq 1$. In all cases we have $\boldsymbol{\eta}^\top d(\boldsymbol{\mu}, \boldsymbol{\lambda}) \geq 1$ for all $\boldsymbol{\lambda} \in B(m,\boldsymbol{\mu})$, hence $\boldsymbol{\eta}$ is feasible. This concludes the proof.

# E   An improved dynamic programming procedure

In this section, we describe a more complex, but more computationally efficient dynamic program than that presented in Section 3, which enables solving $P_{GL}$ more efficiently. Throughout this section we view $G$ as the directed tree $G^{k^\star(\boldsymbol{\mu})}$ rooted at node $k^\star(\boldsymbol{\mu})$.

## E.1   Dynamic programming variables

We describe a dynamic programming procedure to solve the optimization problem

$$\text{minimize}_{\boldsymbol{\lambda}} \ \boldsymbol{\eta}^\top d(\boldsymbol{\mu}, \boldsymbol{\lambda}) \text{ subject to } \boldsymbol{\lambda} \in \cup_{k \neq k^\star(\boldsymbol{\mu})} \tilde{\mathcal{B}}_k(m,\boldsymbol{\mu}) \qquad (\tilde{P}'_{GL})$$

where $\tilde{\mathcal{B}}_k(m,\boldsymbol{\mu}) = \mathcal{B}'_k(m,\boldsymbol{\mu}) \cap D(n,\boldsymbol{\mu})^K$ (refer to Proposition 4 for the definition of $\mathcal{B}'_k(m,\boldsymbol{\mu})$). Consider $\boldsymbol{\lambda}^\star$ the optimal solution to this problem. Then there exists a node $k \neq k^\star(\boldsymbol{\mu})$ such that $\lambda_k^\star = \mu^\star$. With a slight abuse of notation, we denote this node by $k^\star(\boldsymbol{\lambda}^\star)$. We distinguish two cases.

**Case 1.** If $k = k^\star(\boldsymbol{\lambda}^\star) \in \mathcal{N}(\boldsymbol{\mu})$, from Proposition 3, the optimal solution is $\boldsymbol{\lambda}^\star = \boldsymbol{\mu} + (\mu^\star - \mu_k)e^{(k)}$, and the optimal value is $\eta_k d_k(\mu_k, \mu^\star)$.

**Case 2.** If $k = k^\star(\boldsymbol{\lambda}^\star) \notin \mathcal{N}(\boldsymbol{\mu})$, from Proposition 6, the modes of $\boldsymbol{\lambda}^\star$ must be located at $\mathcal{M}(\boldsymbol{\lambda}^\star) = \mathcal{M}(\boldsymbol{\mu}) \cup \{k\} \setminus \{k'\}$, where $k' \in \mathcal{M}(\boldsymbol{\mu}) \setminus \{k^\star(\boldsymbol{\mu})\}$ is the mode of $\boldsymbol{\mu}$ that is not a mode of $\boldsymbol{\lambda}^\star$.

Given a node $\ell$ of $G$, and flags $(z,a,b,c) \in D(n,\boldsymbol{\mu}) \times \{0,1,2\} \times \{0,1\} \times \{0,1\}$ we define $h_\ell(z,a,b,c)$ as the minimal value of $\sum_{j \in \mathcal{D}(\ell) \cup \{\ell\}} \eta_j d_j(\mu_j, \lambda_j)$ where $\mathcal{M}(\boldsymbol{\lambda}^\star) = \mathcal{M}(\boldsymbol{\mu}) \cup \{k^\star(\boldsymbol{\lambda}^\star)\} \setminus \{k'\}$ for some $k' \in \mathcal{M}(\boldsymbol{\mu}) \setminus \{k^\star(\boldsymbol{\mu})\}$, under four constraints:
(i) $\lambda_\ell = z$
(ii) If $a = 0$, $\ell \in \mathcal{M}(\boldsymbol{\lambda}^\star)$, if $a = 1$, $\max_{v \in \mathcal{C}(\ell)} \lambda_v \geq \lambda_\ell$, and if $a = 2$, $\lambda_{p(\ell)} \geq \lambda_\ell$
(iii) $b = \mathbb{1}\{k^\star(\boldsymbol{\lambda}^\star) \in \mathcal{D}(\ell) \cup \{\ell\}\}$
(iv) $c = \mathbb{1}\{k' \in \mathcal{D}(\ell) \cup \{\ell\}\}$
We further define $\boldsymbol{\lambda}^\star(\ell,z,a,b,c)$ as the corresponding optimal solution. Recall that $G^{k^\star(\boldsymbol{\mu})}$ is a tree rooted at $k^\star(\boldsymbol{\mu})$ and that in case 2, $k^\star(\boldsymbol{\mu})$ must be a mode of $\boldsymbol{\lambda}^\star$. Putting the two cases together, the optimal solution to $\tilde{P}'_{GL}$ is

$$\boldsymbol{\lambda}^\star = \begin{cases} \boldsymbol{\mu} + (\mu^\star - \mu_{k^\perp})e^{(k^\perp)} & \text{if } \eta_{k^\perp} d_{k^\perp}(\mu_k^\perp, \mu^\star) \leq h_{k^\star(\boldsymbol{\mu})}(\mu^\star, 0, 1, 1) \\ \boldsymbol{\lambda}^\star(k^\star(\boldsymbol{\mu}), \mu^\star, 0, 1, 1) & \text{otherwise} \end{cases}$$

where $k^\perp \in \arg\min_{k \in \mathcal{N}(\boldsymbol{\mu})} \eta_k d_k(\mu_k, \mu^\star)$.

## E.2   Terminal conditions for leaves

First consider $\ell$ a leaf of $G$. Then five terminal conditions must be satisfied:
(i) Since $a = 1$ requires $\min_{v \in \mathcal{C}(\ell)} \lambda_v \geq \lambda_\ell$, we must have $a \neq 1$
(ii) If $b = 0$ then $\ell \neq k^\star(\boldsymbol{\lambda}^\star)$, so that either $a \neq 0$ or $\ell \in \mathcal{M}(\boldsymbol{\mu})$

(iii) If $b = 1$ then $\ell = k^\star(\boldsymbol{\lambda}^\star)$, so that $a = 0$ and $\ell \notin \mathcal{M}(\boldsymbol{\mu})$
(iv) If $c = 0$ then $\ell \neq k'$, so that either $a = 0$ or $\ell \notin \mathcal{M}(\boldsymbol{\mu})$
(v) If $c = 1$ then $\ell = k'$, so that $a \neq 0$ and $\ell \in \mathcal{M}(\boldsymbol{\mu})$ so that

$$h_\ell(z, a, b, c) = \begin{cases} \eta_\ell d_\ell(\mu_\ell, z) + \sum_{v \in \mathcal{D}(\ell)} h_v(z_v, a_v, b_v, c_v) & \text{if (i) - (v) hold} \\ +\infty & \text{otherwise.} \end{cases}$$

### E.3 Dynamic programming rules for internal nodes

Now consider $\ell$ an internal node of $G$. We wish to compute the value of $h$ recursively using dynamic programming. We first write

$$h_\ell(z, a, b, c) = \eta_\ell d_\ell(\mu_\ell, \lambda_\ell) + \sum_{v \in \mathcal{C}(\ell)} h_v(z_v, a_v, b_v, c_v)$$

where $(z_v, a_v, b_v, c_v)_{v \in \mathcal{C}(\ell)}$ obeys a set of rules:
(i) If $a = 0$ then $\ell$ is a mode of $\lambda$, so $z_v < z$ for all $v \in \mathcal{C}(\ell)$, and $a_v = 2$, because $\lambda_v < \lambda_\ell = \lambda_{p(v)}$
(ii) if $a = 1$ then $\ell$ has a child with higher value, so there must exist at least one $v \in \mathcal{C}(\ell)$ with $z_v \geq z$
(iii) If $b = 0$ then $b_v = 0$ for all $v \in \mathcal{C}(\ell)$, since if the subtree of $\ell$ does not contain $k^\star(\boldsymbol{\lambda}^\star)$, then none of the subtrees of $v$ contain $k^\star(\boldsymbol{\lambda}^\star)$
(iv) If $b = 1$, $a = 0$ and $\ell \notin \mathcal{M}(\boldsymbol{\mu})$, then $\ell = k^\star(\boldsymbol{\lambda}^\star)$, so that none of the subtrees of $v$ contain $k^\star(\boldsymbol{\lambda}^\star)$, i.e., $b_v = 0$ for all $v \in \mathcal{C}(\ell)$.
(v) If $b = 1$ and either $a \in \{1, 2\}$ or $\ell \in \mathcal{M}(\boldsymbol{\mu})$, then $\sum_{v \in \mathcal{C}(\ell)} b_v = 1$, since if the subtree of $\ell$ contains $k^\star(\boldsymbol{\lambda}^\star)$, and $\ell \neq k^\star(\boldsymbol{\lambda}^\star)$ then there must exist exactly one $v$ whose subtree contains $k^\star(\boldsymbol{\lambda}^\star)$
(vi) If $c = 0$ and $\ell \in \mathcal{M}(\boldsymbol{\mu})$ then $a = 0$, since if the subtree of $\ell$ does not contain $k'$ then $\ell \neq k'$, which gives $\ell \in \mathcal{M}(\boldsymbol{\lambda}^\star)$
(vii) If $c = 0$ then $c_v = 0$ for all $v \in \mathcal{C}(\ell)$, since if the subtree of $\ell$ does not contain $k'$ then the subtree of all $v$ does not contain $k'$
(viii) If $c = 1$ then either $a = \{1, 2\}$ and $\ell \in \mathcal{M}(\boldsymbol{\mu})$ and we have $\ell = k'$, which implies $c_v = 0$ for all $v \in \mathcal{D}(\ell)$, or there must exist exactly one $v$ whose subtree contains $k'$, so that $\sum_{v \in \mathcal{D}(\ell)} c_v = 1$
(ix) If $z_v \leq z$ then $a_v = 2$, otherwise $a_v \in \{0, 1\}$.

### E.4 Recursive equations for internal nodes

Based on those rules we compute the value of $h_\ell(z, a, b, c)$ recursively using dynamic programming. To do so we define the following auxiliary functions where, as in Section 4.2, the minima are taken with the implicit constraint $w \in D(n, \boldsymbol{\mu})$:

$$h_\ell^>(z, b, c) = \min_{w > z} \min_{a \in \{0,1\}} h_\ell(w, a, b, c) \,, \ h_\ell^<(z, b, c) = \min_{w < z} h_\ell(w, 2, b, c)$$

$$h_\ell^{\geq}(z, b, c) = \min(h_\ell^>(z, b, c), h_\ell(z, 2, b, c)) \,, \ h_\ell^\star(z, b, c) = \min(h_\ell^<(z, b, c), h_\ell(z, 2, b, c), h_\ell^>(z, b, c))$$

$$h_\ell^\Delta(z, b, c) = |\mathcal{C}(\ell)|^{-1} \min_{v \in \mathcal{C}(\ell)} \left\{ h_v^{\geq}(z, b, c) - h_v^\star(z, b, c) \right\}$$

$$\mathcal{X}_\ell(s_1, s_2) = \left\{ (x_{1,v}, x_{2,v})_{v \in \mathcal{C}(\ell)} \in \{0, 1\}^{2 \times \mathcal{C}(\ell)} : \sum_{v \in \mathcal{C}(\ell)} (x_{1,v}, x_{2,v}) = (s_1, s_2) \right\} \text{ for } (s_1, s_2) \in \mathbb{Z}^2$$

where it is noted that $\mathcal{X}_\ell(s_1, s_2) = \emptyset$ if $\min(s_1, s_2) < 0$.

We have $h_\ell(z, a, b, c) = +\infty$ if any of the three conditions hold:
(i) $a = 0$, $\ell \notin \mathcal{M}(\boldsymbol{\mu})$ and $b = 0$
(ii) $a = 0$, $\ell \notin \mathcal{M}(\boldsymbol{\mu})$, and $z \neq \mu^\star$
(iii) $a \in \{1, 2\}$ and $\ell \in \mathcal{M}(\boldsymbol{\mu})$ and $c = 0$.
Indeed, if $a = 0$ and $\ell \notin \mathcal{M}(\mu)$ we must have $\ell = k^\star(\boldsymbol{\lambda}^\star)$, so that in turn $b = 1$, and $\lambda_\ell = \mu^\star$. Also, if $a \in \{1, 2\}$ and $\ell \in \mathcal{M}(\mu)$ then we must have $\ell = k'$, which imposes $c = 1$.

Otherwise, the value of $h_\ell(z, a, b, c)$ is given by the following recursive equations, where by convention, minimization over an empty set yields $+\infty$:

$$h_\ell(z, 0, b, c) = \eta_\ell d_\ell(\mu_\ell, z) + \min_{(\mathbf{b},\mathbf{c}) \in \mathcal{X}_\ell(b - \mathbf{1}\{\ell \notin \mathcal{M}(\boldsymbol{\mu})\}, c)} \left\{ \sum_{v \in \mathcal{C}(\ell)} h_v^<(z, b_v, c_v) \right\}$$

$$h_\ell(z, 1, b, c) = \eta_\ell d_\ell(\mu_\ell, z) + \min_{(\mathbf{b},\mathbf{c}) \in \mathcal{X}_\ell(b, c - \mathbf{1}\{\ell \in \mathcal{M}(\boldsymbol{\mu})\})} \min_{w \in \mathcal{C}(\ell)} \left\{ h_{\overline{w}}^\geq(z, b_w, c_w) + \sum_{v \in \mathcal{C}(\ell), v \neq w} h_v^\star(z, b_v, c_v) \right\}$$

$$= \eta_\ell d_\ell(\mu_\ell, z) + \min_{(\mathbf{b},\mathbf{c}) \in \mathcal{X}_\ell(b, c - \mathbf{1}\{\ell \in \mathcal{M}(\boldsymbol{\mu})\})} \min_{w \in W_\ell(z)} \left\{ h_{\overline{w}}^\geq(z, b_w, c_w) + \sum_{v \in \mathcal{C}(\ell), v \neq w} h_v^\star(z, b_v, c_v) \right\}$$

$$h_\ell(z, 2, b, c) = \eta_\ell d_\ell(\mu_\ell, z) + \min_{(\mathbf{b},\mathbf{c}) \in \mathcal{X}_\ell(b, c - \mathbf{1}\{\ell \in \mathcal{M}(\boldsymbol{\mu})\})} \left\{ \sum_{v \in \mathcal{C}(\ell)} h_v^\star(z, b_v, c_v) \right\}$$

with

$$W_\ell(z) = \cup_{(b,c) \in \{0,1\}^2} \left\{ \underset{w \in \mathcal{C}(\ell)}{\arg\min} \left[ h_{\overline{w}}^\geq(z, b, c) - h_w^\star(z, b, c) \right] \right\}$$

so that $|W_\ell(z)| \leq 4$ and where we used the fact that

$$\underset{w \in \mathcal{C}(\ell)}{\arg\min} \left\{ h_{\overline{w}}^\geq(z, b_w, c_w) + \sum_{v \in \mathcal{C}(\ell), v \neq w} h_v^\star(z, b_v, c_v) \right\}$$

$$= \underset{w \in \mathcal{C}(\ell)}{\arg\min} \left\{ h_{\overline{w}}^\geq(z, b_w, c_w) - h_w^\star(z, b_w, c_w) + \sum_{v \in \mathcal{C}(\ell)} h_v^\star(z, b_v, c_v) \right\}$$

$$= \underset{w \in \mathcal{C}(\ell)}{\arg\min} \left\{ h_{\overline{w}}^\geq(z, b_w, c_w) - h_w^\star(z, b_w, c_w) \right\} \in W_\ell(z)$$

### E.5 Fast evaluation of recursive equations

We now propose an efficient strategy to compute the minimization problems in the recursive equations. For any function $\phi$, we can minimize $\sum_{v \in \mathcal{C}(\ell)} \phi_v(b_v, c_v)$ over $(\mathbf{b}, \mathbf{c}) \in \mathcal{X}_\ell(s_1, s_2)$ in time and memory $O(|\mathcal{C}(\ell)|)$ for any $(s_1, s_2) \in \{0, 1\}^2$ using the following strategy. If $(s_1, s_2) = (0, 0)$, then trivially

$$\min_{(\mathbf{b},\mathbf{c}) \in \mathcal{X}_\ell(0,0)} \left\{ \sum_{v \in \mathcal{C}(\ell)} \phi_v(b_v, c_v) \right\} = \sum_{v \in \mathcal{C}(\ell)} \phi_v(0, 0)$$

If $(s_1, s_2) = (1, 0)$,

$$\min_{(\mathbf{b},\mathbf{c}) \in \mathcal{X}_\ell(1,0)} \left\{ \sum_{v \in \mathcal{C}(\ell)} \phi_v(b_v, c_v) \right\} = \min_{\mathbf{b} \in \{0,1\}^{\mathcal{C}(\ell)} : \sum_{v \in \mathcal{C}(\ell)} b_v = 1} \left\{ \sum_{v \in \mathcal{C}(\ell)} \phi_v(b_v, 0) \right\}$$

$$= \min_{w \in \mathcal{C}(\ell)} \left\{ \phi_w(1, 0) - \phi_w(0, 0) \right\} + \sum_{v \in \mathcal{C}(\ell)} \phi_v(0, 0)$$

and by symmetry, if $(s_1, s_2) = (0, 1)$,

$$\min_{(\mathbf{b},\mathbf{c}) \in \mathcal{X}_\ell(0,1)} \left\{ \sum_{v \in \mathcal{C}(\ell)} \phi_v(b_v, c_v) \right\} = \min_{w \in \mathcal{C}(\ell)} \left\{ \phi_w(0, 1) - \phi_w(0, 0) \right\} + \sum_{v \in \mathcal{C}(\ell)} \phi_v(0, 0)$$

and finally, if $(s_1, s_2) = (1, 1)$,

$$\min_{(\mathbf{b},\mathbf{c}) \in \mathcal{X}_\ell(1,1)} \left\{ \sum_{v \in \mathcal{C}(\ell)} \phi_v(b_v, c_v) \right\} = \min(\Delta, \Delta') + \sum_{v \in \mathcal{C}(\ell)} \phi_v(0, 0)$$

with

$$\Delta = \min_{w \in \mathcal{C}(\ell)} \left\{ \phi_w(1, 1) - \phi_w(0, 0) \right\}$$

$$\Delta' = \min_{w_1, w_2 \in \mathcal{C}(\ell)^2, w_1 \neq w_2} \left\{ \phi_{w_1}(1, 0) - \phi_{w_1}(0, 0) + \phi_{w_2}(0, 1) - \phi_{w_2}(0, 0) \right\}$$

In all cases, one can compute the minimization in time $O(|\mathcal{C}(\ell)|)$. In particular $\Delta'$ can be computed by realizing that the only pairs $(w_1, w_2)$ that minimize the expression must be either the first or second smallest entry of $\phi_v(1,0) - \phi_v(0,0)$ and $\phi_v(0,1) - \phi_v(0,0)$. We recall that, finding the first and second smallest entry of a vector can be done in time proportional to the vector size, by inspecting each entry at most twice.

## E.6 Retrieving the optimal solution

Once the value of $h_\ell(z, a, b, c)$ has been determined, we can retrieve the optimal solution $\boldsymbol{\lambda}^\star(k^\star(\boldsymbol{\mu}), \mu^\star, 0, 1, 1)$ by retrieving the value of the flags $(z_\ell^\star, a_\ell^\star, b_\ell^\star, c_\ell^\star)$ for all $\ell$. Consider a node $\ell \neq k^\star(\boldsymbol{\mu})$, once its flags $(z_\ell^\star, a_\ell^\star, b_\ell^\star, c_\ell^\star)$ have been computed, we compute the flags of its children as follows.

(i) If $a_\ell^\star = 0$, then

$$(b_v^\star, c_v^\star)_{v \in \mathcal{C}(\ell)} \in \underset{(\mathbf{b},\mathbf{c}) \in \mathcal{X}_\ell(b - \mathbb{1}\{\ell \notin \mathcal{M}(\boldsymbol{\mu})\}, c)}{\arg \min} \left\{ \sum_{v \in \mathcal{C}(\ell)} h_v^<(z_\ell^\star, b_v, c_v) \right\} \quad \text{and}$$

$$z_v^\star \in \arg \min_{z < z_\ell^\star} h_v(z, 2, b_v^\star, c_v^\star).$$

(ii) If $a_\ell^\star = 1$, then

$$(b_v^\star, c_v^\star)_{v \in \mathcal{C}(\ell)} \in \underset{(\mathbf{b},\mathbf{c}) \in \mathcal{X}_\ell(b, c - \mathbb{1}\{\ell \in \mathcal{M}(\boldsymbol{\mu})\})}{\arg \min} \min_{w \in W_\ell(z_\ell^\star)} \left\{ h_w^\geq(z_\ell^\star, b_w, c_w) + \sum_{v \in \mathcal{C}(\ell), v \neq w} h_v^\star(z_\ell^\star, b_v, c_v) \right\},$$

$$z_v^\star = \begin{cases} \arg\min_{z > z_\ell^\star} \min_{a \in \{0,1\}} h_v(z, a, b_v^\star, c_v^\star) & \text{if } v = w_\ell^\star \text{ and } h_v^>(z_\ell^\star, b_v^\star, c_v^\star) = h_v^\geq(z_\ell^\star, b_v^\star, c_v^\star) \\ z_\ell^\star & \text{if } v = w_\ell^\star \text{ and } h_v(z_\ell^\star, 2, b_v^\star, c_v^\star) = h_v^\geq(z_\ell^\star, b_v^\star, c_v^\star) \\ \arg\min_{z > z_\ell^\star} \min_{a \in \{0,1\}} h_v(z, a, b_v^\star, c_v^\star) & \text{if } v \neq w_\ell^\star \text{ and } h_v^>(z_\ell^\star, b_v^\star, c_v^\star) = h_v^\star(z_\ell^\star, b_v^\star, c_v^\star) \\ z_\ell^\star & \text{if } v \neq w_\ell^\star \text{ and } h_v(z_\ell^\star, 2, b_v^\star, c_v^\star) = h_v^\star(z_\ell^\star, b_v^\star, c_v^\star) \\ \arg\min_{z < z_\ell^\star} h_v(z, 2, b_v^\star, c_v^\star) & \text{if } v \neq w_\ell^\star \text{ and } h_v^<(z_\ell^\star, b_v^\star, c_v^\star) = h_v^\star(z_\ell^\star, b_v^\star, c_v^\star) \end{cases}$$

where

$$w_\ell^\star \in \underset{w \in W_\ell(z_\ell^\star)}{\arg \min} \left\{ h_w^\geq(z_\ell^\star, b_w^\star, c_w^\star) - h_w^\star(z_\ell^\star, b_w^\star, c_w^\star) \right\}$$

(iii) If $a_\ell^\star = 2$, then

$$(b_v^\star, c_v^\star)_{v \in \mathcal{C}(\ell)} \in \underset{(\mathbf{b},\mathbf{c}) \in \mathcal{X}_\ell(b, c - \mathbb{1}(\ell \in \mathcal{M}(\boldsymbol{\mu})))}{\arg \min} \left\{ \sum_{v \in \mathcal{C}(\ell)} h_v^\star(z_\ell^\star, b_v, c_v) \right\} \quad \text{and}$$

$$z_v^\star = \begin{cases} \arg\min_{z > z_\ell^\star} \min_{a \in \{0,1\}} h_v(z, a, b_v^\star, c_v^\star) & \text{if } h_v^>(z_\ell^\star, b_v^\star, c_v^\star) = h_v^\star(z_\ell^\star, b_v^\star, c_v^\star) \\ z_\ell^\star & \text{if } h_v(z_\ell^\star, 2, b_v^\star, c_v^\star) = h_v^\star(z_\ell^\star, b_v^\star, c_v^\star) \\ \arg\min_{z < z_\ell^\star} h_v(z, 2, b_v^\star, c_v^\star) & \text{if } h_v^<(z_\ell^\star, b_v^\star, c_v^\star) = h_v^\star(z_\ell^\star, b_v^\star, c_v^\star) \end{cases}$$

Finally, $a_v^\star = 2$ if $z_\ell^\star \geq z_v^\star$ and $a_v^\star \in \arg\min_{a \in \{0,1\}} h_v(z_v^\star, a, b_v^\star, c_v^\star)$ otherwise. In practice, these argmins can be stored during the forward pass (where we compute each $h_\ell(z, a, b, c)$) and do not need to be recomputed.

## E.7 Computational complexity

Recall that $z \in D(\boldsymbol{\mu}, n)$, which is a grid of size $n$. If $h_v(z, a, b, c)$ has been computed for all $(z, a, b, c)$ and all $v \in \mathcal{C}(\ell)$, then one can compute $h_v^>(z, b, c)$, $h_v^=(z, b, c)$, $h_v^<(z, b, c)$, $h_v^\geq(z, b, c)$, $h_v^\star(z, b, c)$ for all $(z, a, b, c)$ and all $v \in \mathcal{C}(\ell)$ in time $O(n|\mathcal{C}(\ell)|)$. In turn, using the fast evaluation strategy explained above, one can compute $h_\ell(z, a, b, c)$ for all $(z, a, b, c)$ in time $O(n|\mathcal{C}(\ell)|)$. Therefore, the total time and memory required to solve $\tilde{P}'_{GL}$ with this strategy is $O(n \sum_\ell |\mathcal{C}(\ell)|) = O(nK)$ since $G$ is a tree.

## E.8 Runtime comparison in practice

The improved dynamic program (DP) runs in time $O(Kn)$, compared to $O(K^2 mn)$ for the procedure described in Section 4.2, which we will refer to as the original DP in what follows. In practice,

however, the improved DP may run slower than the original DP on tree instances with moderate values of $K$. To clarify when one should use each procedure, we report their average runtime over 50 trials on specific tree instances, with $\boldsymbol{\eta}$ uniformly sampled at random in $[0,1]^K$, $\boldsymbol{\mu}$ generated as in Section 6 with $m = |\mathcal{M}(\boldsymbol{\mu})| = 3$, $\sigma = 2$, and a discretization parameter of $n = 100$. We pick a Gaussian divergence: $d_k(\lambda_k, \mu_k) = (\lambda_k - \mu_k)^2/2$ for each $k \in [K]$. We perform two experiments:

(i) We measure runtime on random trees as the number of nodes $K$ increases, with $K \in \{100, 400, 700, 1000, 1300, 1600, 1900\}$.

(ii) We measure runtime on balanced $d$-ary trees (i.e., each node has $d$ children) of a fixed height $h = 3$. We vary the branching factor $d \in \{2, 4, 6, 8, 10, 12\}$, which implicitly varies the number of nodes $K$ from 15 to 1885.

The results are reported in Figure 6, along with $95\%$ confidence intervals using bootstrap. The left panel shows that for random trees, the original DP is faster on average up to $K = 1000$, after which the improved DP generally runs faster. The difference is more pronounced in the right panel, with the average runtime of the original DP increasing much faster with the branching factor $d$ of the balanced tree.

Notably, the original DP exhibits higher runtime variance. This may be explained by an implementation trick we applied to reduce its runtime: specifically, before running the complete dynamic programming subroutine for each $k \notin \mathcal{N}(\mu)$, we check whether $\eta_k d_k(\mu_k, \mu^\star) \geq \min_{\ell \in \mathcal{N}(\mu)} \eta_\ell d_\ell(\mu_\ell, \mu^\star)$. If this holds, the subroutine will not find a parameter with a smaller value than the trivial solution of Proposition 3, hence it is skipped. The number of calls to the subroutine is therefore highly instance-dependent.

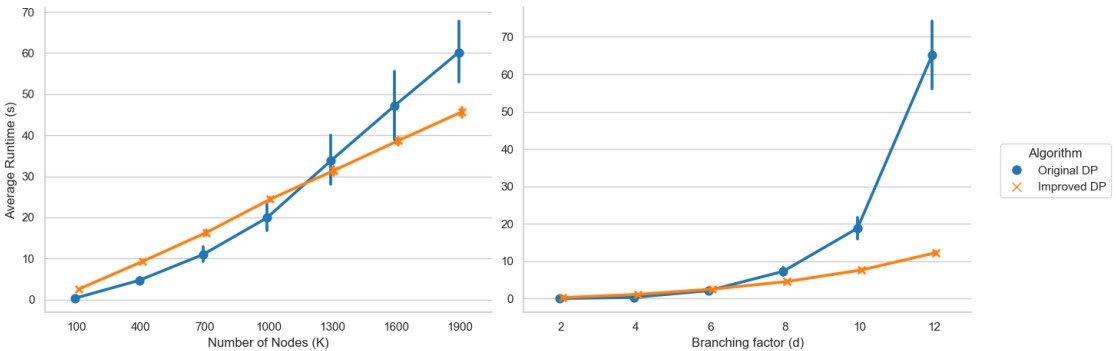

Figure 6: Average runtime of each dynamic program with respect to the number of nodes $K$ or the branching factor $d$.

