# OpenReview forum: "Multimodal Bandits: Regret Lower Bounds and Optimal Algorithms"
_NeurIPS.cc/2025/Conference — NeurIPS 2025 poster_

### Official Review · Reviewer_KMjT · 2025-06-30

**Clarity:** 3
**Significance:** 3
**Originality:** 3
**Rating:** 4
**Confidence:** 3

**Summary:**

This paper studies the problem of stochastic multi-armed multimodal bandit: the algorithm additionally knows a graph $G$, and the hidden vector $\mu$ has at most $m$ modes with respect to this graph (where $\mu_i$ is a mode if $\mu_i > \max_{\text{$j$ adjacent to $i$ on $G$}} \mu_j$). The paper proposes a tractable algorithm to compute the Graves-Lai optimization program with respect to the structured bandit problem, from which one can apply known algorithms to obtain asymptotically optimal regret bound for this problem.

Previous results on solving the Graves-Lai (GL) optimization program for related settings, such as unimodal, relied on local search. This paper showed that local search strategies cannot be used to solve the GL program in the multimodal settings, thus necessitating new algorithmic ideas. The authors showed some structural properties of the optimal solution for the program, thus enabling the calculation of the optimal solution using dynamic programming on a discretized grid, followed by a gradient descent to further approximate the solution. The authors also provided some experimental results to back up the claim that the new GL optimizer + OSSB indeed gives a significantly better regret, compared to previous results based on classical OSSB or using local search to solve the optimization program.

**Questions:**

**Questions**:
1) I believe that Prop 3-6 holds without the tree assumption (from a quick skim through the proofs of them), while I agree that the dynamic programming step absolutely needs the fact that $G$ is a tree, so going beyond the tree case would require a new approach instead of DP. Do the authors have any proposal for such a technique that goes beyond the tree case, or can you show that in fact there is some hardness in solving such optimization problem under the general graph case?

**Comments and Suggestions**:
1) I believe the dynamic programming can be sped up significantly. From Prop 6, the modes of $\lambda^*$ is exactly the modes of $\mu$, but with one modes of $\mu$ removed and one additional mode added (that is not in $N(\mu)$). You then do a double for loop to choose these two modes (time complexity $O(Km)$) before doing the dynamic programming (time complexity $O(Kn)$). I believe you can simply fold the outer loops into the DP by introducing two flags in the DP state: whether you have chosen a mode of $\mu$ to remove in the subtree, and whether you have chosen a new mode to be added in the subtree. This should free the DP of the outer two loops and give you an $O(Kn)$ DP (and an $O(Knt)$ algorithm overall).
2) For completeness, I think the authors should still include a self-contained argument for Prop 1, maybe by simply stating the previous result and explaining why Prop 1 is a direct consequence of it.

**Ethical Concerns:**

["NO or VERY MINOR ethics concerns only"]

**Final Justification:**

I have read the rebuttals and decided to keep my score.

**Limitations:**

yes

**Quality:**

3

**Strengths And Weaknesses:**

**Strengths**:
- The paper is well-written, and I believe the structural results on the optimal solution under this setting (as discussed in the paper) is important while also technically challenging.
- The algorithm given by this paper seems to be reasonable fast (although I believe it can be sped up, see comments below), and I believe the experiments do back up their results that this new algorithm gives a considerably better regret bound, compared to previous results.

**Weaknesses**:
- The biggest weakness of this paper would be the assumption that $G$ is a tree. This is not an assumption found in previous multimodal references, and I am not entirely sure why the authors impose such an assumption other than technical difficulties of not imposing so.
- I am also perplexed by the runtime plot. You shouldn't need a runtime plot to prove that your algorithm have some time complexity; it would have been better to include such a plot to compare your algorithm's runtime against previous results.

---

> ### Author Rebuttal · Authors · 2025-07-30
>
> Many thanks for your positive words and valuable feedback. We address your concerns below.
>
> * **On the tree assumption**. As you pointed out, Propositions 3-6 do not require the tree assumption. We note that our results on local search strategies and peaked reward functions (Section 5) do not require the tree assumption either. It is however key to apply the dynamic programming procedure which enables us to solve the subproblems, and achieving optimality without this assumption would require a more involved procedure. Although we were unable to prove this despite honest attempts, we believe that if the graph $G$ is not assumed to be a tree, then the Graves-Lai optimization problem would not be solvable in polynomial time in the worst case. Some intuition for this stems from Proposition 1, which shows that the set of confusing parameters has a highly complex structure.
>
> * **Runtime plot**. We agree that the plot is not necessary to prove that our algorithm has a time complexity quadratic in the number of arms. We included it to give a better understanding of the runtime in practice on a standard laptop.
>
> * **Runtime speedup**. Thank you for this suggestion. We believe that this strategy would not work for our specific problem.  In our understanding, the core of your suggestion is to augment the DP state with two binary flags to track whether a new mode has already been added and a mode of $\mu$ has already been removed. However, the values of the added and removed modes should also be remembered: the DP recursion must enforce mode constraints specific to which mode $k$ is being added and which mode $k'$ is being removed. Encoding these values in the states would multiply the number of states by $O(Km)$, thus recovering the
> $O(Km)$ enumeration cost we currently pay in the outer loops. Of course, if we have misinterpreted your argument, we would be grateful for any further details or clarification.
>
> * **Self-contained argument for Proposition 1**.  Proposition 1 follows from Theorem 1 in [1] by setting $\Theta=F_{\le m}$ and noting that if the relative entropy $d_k(\mu_{k^{\star}(\mu)}, \lambda_{k^{\star}(\mu)})=0$, then by our Assumption $1$, $\mu_{k^{\star}(\mu)}=\lambda_{k^{\star}(\mu)}$. We will include this argument for completeness in the revised version.
>
> [1] Richard Combes, Stefan Magureanu, and Alexandre Proutiere. Minimal exploration in structured
> stochastic bandits. In Advances in Neural Information Processing Systems, 2017.304

---

> > ### Comment · Reviewer_KMjT · 2025-08-04
> >
> > Thank you for the response. I am happy with points 1, 2, and 4 above.
> >
> > For point 3, I still think a speed up is possible, although I agree that using only 2 binary flags is probably not enough, and one would probably need to introduce more flags. Here's a sketch of my argument.
> >
> > Let's shorthand $k_\lambda = k^\star(\lambda)$ and $k_\mu = k^\star(\mu)$. Then,
> > - Prop. 1 gives $k_\lambda \neq k_\mu$ (there's also $\lambda_{k_\lambda} = \mu_{k_\mu}$, but this is superceded by Prop. 4).
> > - Prop. 3 gives $k_\lambda \notin \mathcal{N}(\mu)$.
> > - Prop. 4 gives $\lambda_{k_\lambda} = \lambda_{k_\mu} = \mu_{k_\mu}$, and $\lambda \in [\mu_{\star}, \mu^{\star}]^K$.
> > - Prop. 6 gives $\mathcal{M}(\lambda) = \mathcal{M}(\mu) + k_\lambda - k'$ with some $k' \in \mathcal{M}(\mu)$.
> >
> > Then I believe the following DP is sufficient: let $dp_\ell(z, a, b, c)$ be the minimal obtainable value of $\sum_{j \in subtree(\ell)} \eta_j d(\mu_j, \lambda_j)$ subject to the following condition:
> > 1. $\lambda_\ell = z$.
> > 2. $a \in \{0, 1, 2\}$.
> >    - If $a = 0$, we promise that $\ell$ is a mode of $\lambda$ (meaning that $\lambda_\ell$ is at least $\lambda$ of all immediate children, and we need to enforce that $\lambda_\ell \ge \lambda_p(\ell)$).
> >    - If $a = 1$, we promise that $\ell$ is NOT a mode of $\lambda$, and some immediate child $v$ some $\ell$ already has $\lambda_v > \lambda_\ell$, so we don't need to enforce a promise on $p(\ell)$.
> >    - If $a = 2$, we promise that $\ell$ is NOT a mode of $\lambda$, but $\lambda_\ell$ is at least $\lambda$ of all immediately children. This means we need to enforce a condition $\lambda_\ell < \lambda_{p(\ell)}$.
> > 3. $b \in \{0, 1\}$. If $b = 1$, $k_\lambda \in subtree(\ell)$; otherwise, $k_\lambda \notin subtree(\ell)$.
> > 4. $c \in \{0, 1\}$. If $c = 1$, $k' \in subtree(\ell)$; otherwise, $k' \notin subtree(\ell)$.
> >
> > One can observe that once you impose whether $\ell$ should be a mode or not, the identity of $\ell$ as $k_\lambda$, $k'$, or neither is immediately known. This means you can impose the conditions from the propositions locally at node $\ell$, and do not have to store the value of $\lambda_\ell$ one level up.
> >
> > Let me know if this has any errors.

---

> ### Author Response · Authors · 2025-08-06
>
> Many thanks for your insightful suggestion. We are enthusiastic about your idea and we found the logic to be generally sound. We describe the strategy in greater detail below (with some minor modifications). In particular, we:
>
> 1) Describe the DP states and constraints.
>
> 2) Derive the DP recursion equations, and show that a naive implementation would still require $O(K^2mn)$ runtime, as our current method.
>
> 3) Suggest a more intricate implementation with $O(Kn)$ runtime, as you initially stated.
>
> ---
>
> ### 1. DP States and constraints.
>
> Consider the directed tree rooted at $k_{\mu}$, and  for any node $\ell$, let $f_{\ell}(z,a,b,c)$ the minimal obtainable value of $\sum_{j \in \mathcal{D}(\ell) \cup \{\ell\}}\eta_{j}d(\mu_{j},\lambda_{j})$ under the constraints $\lambda_{\ell}=z$ and the following flag constraints on $\lambda$:
> - $a \in \\{0,1,2\\}$ encodes a constraint on the modality of $\ell$:
>    - $a=0$: $\ell \in M(\lambda)$, i.e. $\lambda_{\ell} > \lambda_{p(\ell)}$ and $\lambda_{\ell} > \lambda_v$ $\forall v \in \mathcal{C}(\ell)$.
>   - $a=1$: $\exists  v \in \mathcal{C}(\ell)$, $\lambda_{v} > \lambda_{\ell}$.
>   - $a=2$: $\lambda_{p(\ell)} > \lambda_{\ell}$.
>
> -  $b=1$ if the new mode $k_{\lambda} \notin N(\mu)$ is in the subtree rooted at $\ell$, $b=0$ otherwise
>  - $c=1$ if the removed mode $k' \in M(\mu)$ is in the subtree rooted at $\ell$, $c=0$ otherwise
>
>  Assuming that $|M(\mu)|=m,$ by Proposition 3, the value of the discretized $P_{GL}$ optimization problem is equal to
> $\min(d,f_{k_{\mu}}(\mu^{\star},0,1,1))$  where $d= \min_{k \in N(\mu) \setminus \\{k_{\mu}\\}}\eta_kd(\mu_k,\mu^{\star})$.
>
> For a fixed $\ell \in [K]$ and states $z,a,b,c$,  let $g_{\ell}(z,a,b,c)$ the minimal value of $\sum_{j \in \mathcal{D}(\ell)}\eta_jd(\mu_j,\lambda_j)$ over all valid child state configurations, so that \begin{align*}f_{\ell}(z,a,b,c)=\eta_{\ell}d(\mu_{\ell},z)+g_{\ell}(z,a,b,c)
> \end{align*}
> where $g_{\ell}(z,a,b,c)=\min_{\text{valid child states}}\sum_{v \in \mathcal{C}(\ell)}\sum_{j \in \mathcal{D}(v) \cup \\{v\\}}\eta_{j}d(\mu_{j},\lambda_{j}).$
>
> **Enforcing the constraint $\lambda_{k_\lambda}=\mu^{\star}$ locally.**
> As you noted, the value of $a$ is enough to identify whether $\ell=k_{\lambda}$ or $\ell=k'$:
>
> (i) If $a=0$ and $\ell \notin M(\mu)$, then $\ell$ is the new mode $k_{\lambda}$ and we must enforce $\lambda_{\ell}=\mu^{\star}.$
>
> (ii) If $a \ne 0$ and $\ell \in M(\mu),$ then $\ell$ is the removed mode $k'$.
>
> (iii) In all other cases,  $\ell$ is neither the new mode nor the removed mode.
>
> ----
>
> ### 2. DP recursion equations & naive implementation.
>
> We now relate $g_{\ell}(z,a,b,c)$ to the $f_{v}$ for $v \in \mathcal{C}(\ell).$ There are many cases to distinguish; we describe them below.
>
> **Case $a=2$, $b=c=1$.** $a=2$ places no constraints on the values of $\lambda_v$ for $v \in \mathcal{C}(\ell)$.  Let $\textbf{b}=(b_v)$ and $\textbf{c}=(b_v)$ be binary vectors indicating the flag assignments for the children of $\ell$. $b=1$ imposes that $b_v=1$ for exactly one $v \in \mathcal{C}(\ell)$. Similarly, $c=1$ imposes that $c_{v'}=1$ for exactly one $v' \in \mathcal{C}(\ell)$. Importantly, this $v'$ must contain a mode of $\mu$ in its subtree, i.e. $\text{subtree}(v') \cap M(\mu) \ne \emptyset.$  We define a constraint set $A_{\ell}$ as:
>
> $$A_{\ell} = \\{ (\textbf{b}, \textbf{c}) \in \\{0,1\\}^{2|\mathcal{C}(\ell)|} : \sum_{v \in \mathcal{C}(\ell)} b_v = 1,  \sum_{v \in \mathcal{C}(\ell)} c_v = 1,  \text{and if } c_v=1 \text{ then } \text{subtree}(v) \cap M(\mu) \neq \emptyset \\}$$
>
> Then, $$g_{\ell}(z,2,1,1)=\min_{(\textbf{b},\textbf{c}) \in A_{\ell} }  \sum_{v \in \mathcal{C}(\ell)} f_v^{\star}(b_{v},c_{v}) $$  where $$f_v^{\star}(b_v,c_v)=\min_{z_v \in D(n,\mu)} \min_{a_v=0,1,2}  f_{v}(z_v,a_v,b_v,c_v).$$
>
> Note that $|A_{\ell}| \le |\mathcal{C}(\ell)|\min(m,|\mathcal{C}(\ell)|) \le m| \mathcal{C}(\ell)|$. Assuming all the values $f_{v}(z_v,a_v,b_v,c_v)$ and $f_v^{\star}(b_v,c_v)$ are known, we can naively compute $g_{\ell}(z,2,1,1)$ with $O(|A_{\ell}||\mathcal{C}(\ell)|)=O(m|\mathcal{C}(\ell)|^2)$ operations by calculating  $\sum_{v \in \mathcal{C}(\ell)} f_v^{\star}(b_{v},c_{v})$ for each $(\textbf{b},\textbf{c}) \in A_{\ell}$. Computing $f_{\ell}(z,2,1,1)$ for all $z \in D(n,\mu)$ and $f_{\ell}^{\star}(1,1)$ can then be done with $O(|D(n,\mu)|m|\mathcal{C}(\ell)|^2)=O(mn|\mathcal{C}(\ell)|^2)$ operations.
>
> **Case $a=2$, $b=c=0$.** Since $b=c=0$ forces $b_v=c_v=0$ for all $v \in \mathcal{C}(\ell)$, we directly have $$g_{\ell}(z,2,0,0)=\displaystyle \sum_{v \in \mathcal{C}(\ell)} \min_{z_v \in D(n,\mu)} \min_{a_v=0,1,2}  f_{v}(z_v,a_v,0,0) $$ and computing $f_{\ell}(z,2,0,0)$, $f^{\star}_{\ell}(0,0)$ requires $O(n|\mathcal{C}(\ell)|)$ operations.
>
> The other subcases are treated similarly. Generally, each recursion step for cases where $a=2$ requires at most $O(mn|\mathcal{C}(\ell)|^2)$ operations with the naive computation method.

---

> ### Author Response · Authors · 2025-08-06
>
> ### 2. DP recursion equations & naive implementation (continued).
>
> **Case $a=1$.** There must be a child $v$ such that $\lambda_v > \lambda_{\ell}$. For example, if $b=c=0$, the children cost is $$g_{\ell}(z,1,0,0)=\displaystyle \sum_{v \in \mathcal{C}(\ell)} \min_{z_v \in D(n,\mu)} \min_{a_v=0,1,2}  f_{v}(z_v,a_v,0,0) + \min_{v \in \mathcal{C}(\ell)} h_v(z)$$ where $$h_v(z)=\min_{z_v \in D(n,\mu), z_v > z} \min_{a_v=0,1,2}  f_{v}(z_v,a_v,0,0)-\min_{z_v \in D(n,\mu)} \min_{a_v=0,1,2}  f_{v}(z_v,a_v,0,0)$$ and the DP recursion can be similarly derived for the other possible values of $b$ and $c$, and in all cases, each recursion step requires at most $O(mn|\mathcal{C}(\ell)|^2)$ operations with the naive method.
>
> **Case $a=0$.** There, the constraint $\lambda_{\ell} > \lambda_v$ for all $v \in \mathcal{C}(\ell)$ must be enforced. Additionally, if $\ell \notin M(\mu)$, we must enforce $\lambda_{\ell}=\mu^{\star}$. If $b=c=0,$ the children cost becomes $$g_{\ell}(z,0,0,0)=\displaystyle \sum_{v \in \mathcal{C}(\ell)} \min_{z_v < z} \min_{a_v=0,1,2}  f_{v}(z_v,a_v,0,0) + \infty \textbf{1}\\{\ell \notin M(\mu) \,\wedge \, z \ne \mu^{\star}\\}$$ where the infinite term is added to enforce $\lambda_{k_\lambda}=\mu^{\star}$.
> The recursion formula is derived similarly for the other values of $b$ and $c$, and in all cases, each recursion step requires at most $O(mn|\mathcal{C}(\ell)|^2)$ operations.
>
> Overall, this new DP can be run in $$O(\sum_{\ell \in [K]}mn|\mathcal{C}(\ell)|^2)$$ time, which matches the  $O(K^2mn)$ complexity of our current simpler DP in the worst case.
>
> ---
>
> ### 3. Implementation with $O(Kn)$ runtime.
>
> We illustrate the idea with the case $a=2,b=c=1$. To improve the runtime, we can simplify the expression of the children cost $g_{\ell}(z,2,1,1)=\min_{(\textbf{b},\textbf{c}) \in A_{\ell} }  \sum_{v \in \mathcal{C}(\ell)} f_v^{\star}(b_{v},c_{v}) $ as follows. There are two cases :
>
>  i) $(b_v,c_v)=(1,1)$ for the same $v \in \mathcal{C}(\ell)$, and every other child has these flags as $0$.
>
> ii) $(b_v,c_v)=(1,0)$ and $(b_v',c_v')=(0,1)$ for $v \ne v',$ every other child has these flags as $0$. Consider $S_0=\sum_{v \in \mathcal{C}(\ell)}f_v^{\star}(0,0).$ If we set
> \begin{align*}
> \Delta_{v}^{(1,1)} = f_v^{\star}(1,1) - f_v^{\star}(0,0),\\
> \Delta_{v}^{(1,0)} = f_v^{\star}(1,0) - f_v^{\star}(0,0),\\
> \Delta_{v}^{(0,1)} = f_v^{\star}(0,1) - f_v^{\star}(0,0),
> \end{align*}
>  we can write
> \begin{align*}\min_{(b_v,c_v)\in A_{\ell}}
> \sum_{v\in\mathcal{C}(\ell)}f_v^{\star}(b_v,c_v)=S_0+\min\\{\min_{v \in \mathcal{C}(\ell)}(\Delta_{v}^{(1,1)}),\min_{i \ne j \in \mathcal{C}(\ell)}(\Delta_{i}^{(1,0)}+\Delta_{j}^{(0,1)})\\}
> =S_0+\min\\{M_{11},M_{10,01}\\}\end{align*} with $M_{11} :=\min_{v \in \mathcal{C}(\ell)}\Delta_{v}^{(1,1)}$, $M_{10,01}
> := \min_{i \ne j \in \mathcal{C}(\ell)}(\Delta_{i}^{(1,0)}+\Delta_{j}^{(0,1)})$.
> Assuming that for each child $v$ the values
> \begin{align*}
> f_v^{\star}(0,0),\\
> f_v^{\star}(1,0),\\
> f_v^{\star}(0,1),\\
> f_v^{\star}(1,1),
> \end{align*} are known, in one $O(|\mathcal{C}(\ell)|)$ pass over $v\in\mathcal{C}(\ell)$ we can compute $S_0,M_{11}$
> and identify the top two children for each of the lists $\{\Delta_{v}^{(1,0)}\}$ and $\{\Delta_{v}^{(0,1)}\}$:
>
> \begin{align*}
> v_{10}^{(1)} = \arg\min_v \Delta_{v}^{(1,0)},  D_{10}^{(1)} = \min_v \Delta_{v}^{(1,0)},\\
> v_{10}^{(2)} = \arg\min_{v\ne v_{10}^{(1)}} \Delta_{v}^{(1,0)},  D_{10}^{(2)} &= \min_{v\ne v_{10}^{(1)}} \Delta_{v}^{(1,0)},\\
> v_{01}^{(1)} = \arg\min_v \Delta_{v}^{(0,1)},  D_{01}^{(1)} = \min_v \Delta_{v}^{(0,1)},\\
> v_{01}^{(2)} = \arg\min_{v\ne v_{01}^{(1)}} \Delta_{v}^{(0,1)}, & D_{01}^{(2)} = \min_{v\ne v_{01}^{(1)}} \Delta_{v}^{(0,1)}.
> \end{align*}
> The split‐flags term $M_{10,01}$ is then $D_{10}^{(1)} + D_{01}^{(1)}$ if $v_{10}^{(1)} \neq v_{01}^{(1)}$, and $\min\\{D_{10}^{(1)} + D_{01}^{(2)},D_{10}^{(2)} + D_{01}^{(1)}\\}$ if $v_{10}^{(1)} = v_{01}^{(1)}$.
>
> Finally,
> $g_{\ell}(z,2,1,1)
> = S_0 +\min\\{M_{11},M_{10,01}\\}$ can be computed in $O(|\mathcal{C}(\ell)|)$ time if all the values $f_{v}(z_v,a_v,b_v,c_v)$ and $f_v^{\star}(b_v,c_v)$ are known. Consequently,  computing $f_{\ell}(z,2,1,1)$ and $f_{\ell}^{\star}(1,1)$  requires $O(|D(n,\mu)||\mathcal{C}(\ell)|)=O(n|\mathcal{C}(\ell)|)$ operations. This upper bound holds for all the other cases as well. The runtime complexity of the overall DP is thus $$O(\sum_{\ell \in [K]}n|\mathcal{C}(\ell)|)=O(Kn),$$ as you initially suggested.
>
> In the camera-ready version, we propose to include both the current paper DP (which is easier to digest), and your more intricate DP with faster runtime.

---

> > ### Comment · Reviewer_KMjT · 2025-08-08
> >
> > Thank you for the details. I have made a pass through them and I believe the implementation for the $O(Kn)$ DP above is sound. As per your proposal, I would suggest including the easier-to-understand $O(K^2mn)$ DP for clarity of the idea, alongside the improved version.

---

### Official Review · Reviewer_Wyp1 · 2025-07-03

**Clarity:** 2
**Significance:** 2
**Originality:** 2
**Rating:** 4
**Confidence:** 3

**Summary:**

This paper considers a multi armed bandit where the arm set is encoded by some graph $G$. An arm is a mode, if its expected reward is greater than all it's neighbors. For some $m>0$ the arm set is assumed to contain $m$ modes. The objective of the learner is to then minimise cumulative regret as in the standard multi armed bandit. The paper focuses on the asymptotic setting, that is as $T\rightarrow \infty$ Optimal algorithms exist, which require one to solve the Graves-Lai optimization problem, which can be computationally challenging in practice. In this paper, the authors propose a computationally tractable method of solving the  Graves-Lai optimization problem, in the case where the graph $G$ is assumed to be a tree. The authors then go on to demonstrate that local search algorithms, that is algorithms only exploring arms in the neighbourhood of the set of modes, cannot be optimal.

**Questions:**

As stated above, regarding the statement that the IMED-MB algorithm from [18 ] is assumed to be a local search algorithm, therefore invalidating the claim made in [18] that IMED-MB is asymptotically optimal. I do not see how this is the case, as the  IMED-MB algorithm has the potential to conduct second order exploration across it's running time, where it is able to explore arms outside the neighbourhood of the modes. I would be grateful to the authors if they could clarify this point. It would be helpful to formally define local search algorithms in the paper, and then explicitly show why IMED-MB is a local search algorithm. Even better, would be to highlight the mistake in the proof of theorem 2 in [18], although I appreciate this may be time consuming.

It would be interesting to consider more complex graphs in the experiments. Also, what was the motivation for the specific choice of $\mu_k$ in the experiments?

**Ethical Concerns:**

["NO or VERY MINOR ethics concerns only"]

**Final Justification:**

My main concern with this paper stemmed from my misunderstanding of IMED-MB algorithm, I now agree that it is indeed a local search strategy and thank the authors for taking the time to explain this to me.  As a result I am happy to raise my score. I share some of the other reviewers concerns that the theoretical contribution is limited. Also, reliance on the assumption of tree structure, without any justification, is a significant limitation. In my opinion, a proof supporting the authors intuition, that if the graph is not assumed to be a tree, then the Graves-Lai optimization problem would not be solvable in polynomial time, would greatly benefit the paper.

**Limitations:**

yes

**Quality:**

3

**Strengths And Weaknesses:**

I have two main concerns with the paper. The first is that the assumption that the graph $G$ is a tree seems quite restrictive and reduces the problem to a very specific instance of multi modal bandits. Furthermore, the algorithm proposed by the authors is very complex. For me the combination of these two points reduces the contribution of the paper. The tree constraint is not required in [18], where the authors also propose a computationally tractable algorithm for multi modal bandits.

This brings me to my second concern, regarding the statement that the IMED-MB algorithm from [18 ] is assumed to be a local search algorithm, therefore invalidating the claim made in [18] that IMED-MB is asymptotically optimal. I do not see how this is the case, as the  IMED-MB algorithm has the potential to conduct second order exploration across it's running time, where it is able to explore arms outside the neighborhood of the modes. I would be grateful to the authors if they could clarify this point. It would be helpful to formally define local search algorithms in the paper, and then explicitly show why IMED-MB is a local search algorithm. Even better, would be to highlight the mistake in the proof of theorem 2 in [18] although I appreciate this may be time consuming. At the moment I have tentatively recommended to reject the paper, however, if the authors can clarify the above point I would be open to increasing my score.

---

> ### Author Rebuttal · Authors · 2025-07-30
>
> Thank you for your insightful remarks. We address your concerns below.
>
> * **Complexity of the algorithm and tree assumption**. We believe that the complexity of our procedure is unavoidable to achieve optimality, as solving the Graves-Lai problem in the multimodal case requires careful handling of non-trivial confusing parameters. As we claim in the paper, our results entail that the simpler IMED-MB algorithm from [18] cannot be optimal.
>
>     The tree assumption is key to apply the dynamic programming procedure which enables us to solve the subproblems. We believe that achieving optimality without this assumption would require a more involved procedure. Although we were unable to prove this despite honest attempts, we believe that if the graph $G$ is not assumed to be a tree, then the Graves-Lai optimization problem would not be solvable in polynomial time in the worst case. Some intuition for this stems from Proposition 1, which shows that the set of confusing parameters has a highly complex structure.
>
>     Furthermore, we note that our results on local search strategies and peaked reward functions (Section 5) do not require the tree assumption. Thus, our Proposition 11 shows that for general graphs and peaked reward functions, one may use a simpler local search algorithm, at the cost of being a constant factor away from optimal.
>
> * **Comparison with IMED-MB.** Thank you for bringing up this concern. Let us first clarify our definition of a local search strategy. Informally, an algorithm follows a local search strategy if it asymptotically explores the arms outsides of the modes neighborhood $N(\mu)$ a *sub-logarithmic* amount of time. More formally, letting $N_k(T)$ be the number of times arm $k$ is selected by some algorithm up to time $T$, we define $$\eta_k=\lim_{T \to \infty} \frac{\mathbb{E}[N_k(T)]}{\log{T}}$$ (assuming that the limit exists for simplicity, otherwise consider $\lim\inf$ and $\lim\sup$) so that if $\eta_k >0$, the algorithm asymptotically selects arm $k$ $\eta_k\log{T}+o(\log{T})$ times. As stated in Definition 1, we say that the algorithm follows a local search strategy if $\eta_k=0$ when $k \notin N(\mu)$.
>
>     Although IMED-MB allows second-order exploration, it is claimed in Theorem 2 of [18] that $$\lim_{T \to \infty} \frac{\mathbb{E}[N_k(T)]}{\log{T}} \le 0$$ for every arm $k$ outside of the neighborhood of the modes of $\mu$. This means that IMED-MB falls under the umbrella of local search strategies. We refer to Appendix C.2, where this is explained in more detail. We will emphasize what $\eta_k$ represents in our Definition 1 of local search strategies in the revised version of the paper.
>
> * **Choice of reward vector in the experiments**.  Our choice of $\mu_k$ as a mixture of exponential functions allows us to easily generate a multimodal reward vector with a set of modes and an optimal arm chosen in advance. Tuning $\sigma$ also allows us to control how "peaked" the reward vector is around its modes. Another possibility would have been to choose $\mu_k$ as a mixture of Gaussian functions (by squaring the distance between arms $j$ and $k$ in the expression of $\mu_k$), but this tends to create easier reward instances with very sharp peaks. We propose to perform additional experiments with more complex tree structures in the revised version.

---

> > ### Comment · Reviewer_Wyp1 · 2025-08-06
> >
> > Thank you for your detailed response to my review and for clarifying my misunderstanding of the IMED MB and local search strategies. I intend to raise my score.

---

### Official Review · Reviewer_PpQT · 2025-07-03

**Clarity:** 4
**Significance:** 3
**Originality:** 3
**Rating:** 4
**Confidence:** 2

**Summary:**

This paper addresses stochastic multi-armed bandits where the expected reward function is multimodal with at most $m$ modes on a tree graph. The authors provide the first known computationally tractable algorithm for solving the Graves-Lai optimization problem in this setting, which enables asymptotically optimal algorithms. The key technical contribution is a sophisticated algorithm combining discretization, dynamic programming, and projected subgradient descent to navigate the non-convex constraint set. The paper also proves that local search strategies, used in prior work, can be arbitrarily suboptimal, establishing the necessity of solving the full Graves-Lai problem.

**Questions:**

1. The algorithm has time complexity $O(K^2mnt)$. How does it perform in practice for large $K$ (say, $K > 1000$)? Have you considered approximate versions that could scale better while maintaining theoretical guarantees?

2. The DP decomposition leverages the acyclic nature of trees; cycles destroy the neat top-down factorisation used to solve the semi-infinite LP in $O(K^2)$.  For general graphs, even bounded-treewidth cases introduce exponentially many confusing parameters and break the current structural lemmas.  What are the main technical barriers to extending this to general graphs? Is the problem fundamentally harder, or is this mainly for technical convenience?

**Ethical Concerns:**

["NO or VERY MINOR ethics concerns only"]

**Limitations:**

yes

**Paper Formatting Concerns:**

none.

**Quality:**

3

**Strengths And Weaknesses:**

### Strengths:
- **Clear problem formulation**: Well-motivated extension of unimodal bandits with clean mathematical setup.
- **Solid theoretical contribution**: First tractable algorithm for Graves-Lai optimization in multimodal bandits, solving an important open problem. This work proves that local search can be arbitrarily suboptimal, justifying the need for more sophisticated approaches. Then the creative combination of discretization, dynamic programming, and optimization technique handling the non-convex constraint set. Complete proofs with careful handling of technical details, including time-dependent learning rates.


### Weaknesses:
- **Limited experimental validation**: Only synthetic data on small instances; no comparison with IMED-MB or evaluation on real applications
- **Computational concerns**: $O(K²mnt)$ complexity not thoroughly analyzed; unclear when the approach becomes impractical.

---

> ### Author Rebuttal · Authors · 2025-07-30
>
> Thank you for your review. We appreciate that you considered our work clear, well-motivated, creative and solid theoretically. We address your questions/concerns below.
>
> * **Numerical experiments**. In the revised version, we propose to perform additional experiments that more closely align with real-world scenarios by considering reward instances encountered in applications such as dynamic pricing (e.g. those in [15] and [24])
>
> * **Time complexity**. On the synthetic example considered in our experiments, our algorithm ran on a laptop in about five minutes for $K=70$ arms, $m=2$ modes, $n=100$ discretization points and $t=100$ iterations of subgradient descent per step (Figure 2(a)). Given the quadratic dependency in $K$ of the runtime complexity, the algorithm would run in about a day for $K=1000$.
>
>   To reduce the runtime in practice, one may reduce the number of discretization points and/or the number of iterations of subgradient descent per step, but this comes at the cost at worsened guarantees, that are made explicit in our Theorem 2. Additionally, if it is known that the reward function is peaked, our Proposition 11 shows that a much simpler local search algorithm that disregards the non-trivial confusing parameters can be used, although it will be a constant factor from optimal.
>
> * **Tree assumption and generalization to general graphs**. As you correctly pointed out, our current procedure heavily relies on this structural assumption; without it, our dynamic programming scheme would not apply. We believe that achieving optimality without this assumption would require a more involved procedure. Although we were unable to prove this despite honest attempts, we believe that if the graph $G$ is not assumed to be a tree, then the Graves-Lai optimization problem would not be solvable in polynomial time in the worst case. Some intuition for this stems from Proposition 1, which shows that the set of confusing parameters has a highly complex structure.
>
>
> [15] Kanishka Misra, Eric M. Schwartz, and Jacob Abernethy. Dynamic online pricing with incomplete information using multiarmed bandit experiments. Marketing Science, 38(2):226–333 252, 2019
>
> [24] Yining Wang, Boxiao Chen, and David Simchi-Levi. Multimodal dynamic pricing. Manage. Sci., 67(10):6136–6152, October 2021

---

> > ### Comment · Reviewer_PpQT · 2025-08-07
> >
> > Thank you for your response, I will maintain my score.

---

### Official Review · Reviewer_jMyB · 2025-07-03

**Clarity:** 2
**Significance:** 3
**Originality:** 3
**Rating:** 4
**Confidence:** 3

**Summary:**

This paper addresses the stochastic multi-armed bandit (MAB) problem under a multimodal reward structure, where the expected reward function has at most $m$ modes with respect to a known graph $G$. The key contribution is the first computationally tractable algorithm for solving the Graves-Lai optimization problem in this setting, which is essential for implementing asymptotically optimal bandit algorithms. The authors:

1. Prove that local search strategies are suboptimal in general multimodal settings.
2. Develop a dynamic programming and subgradient-based method to solve the Graves-Lai problem efficiently.
3. Provide theoretical guarantees and empirical validation of their approach.

**Questions:**

Refer to the weaknesses.

**Ethical Concerns:**

["NO or VERY MINOR ethics concerns only"]

**Final Justification:**

Based on the rebuttal, I have decided to maintain my score.

**Limitations:**

Refer to the weaknesses.

**Quality:**

3

**Strengths And Weaknesses:**

**Strengths:**

1. The problem is relevant.
2. The contribution is significant. It's primarily driven by the mathematical analysis that seems robust. However, I did not check it in great detail.

**Weaknesses:**

1. Solving discretized subproblems using DP might be very expensive depending on the discretization parameter.
2. The numerical experiments are convincing. But, they are limited to synthetic data on line graphs with Gaussian rewards. There is no exploration of performance on more complex graph structures (e.g., trees with varying degrees) or real-world-inspired settings. Can you include some real-world scenarios/datasets?
3. The paper is dense and highly technical. Some sections (e.g., the dynamic programming recursion) could benefit from more intuitive explanations or diagrams. The notation is heavy and may be difficult for readers unfamiliar with structured bandits or Graves-Lai theory.

---

> ### Author Rebuttal · Authors · 2025-07-30
>
> Thank you for your positive review and your remarks. We address your concerns below.
>
> * **Cost of solving subproblems with DP**. In our opinion, the linear dependency in the discretization parameter of the runtime complexity of the algorithm is reasonable; more specifically, it scales as $O(K^2mnt)$, where $K$ is the number of arms, $m$ the number of modes, $n$ is the discretization parameter, and $t$ the iterations of subgradient descent per step (Theorem 2).
>
> * **Numerical experiments**. We will be able to perform additional experiments with more complex tree structures or reward distributions in the revised version. To more closely align with real-world scenarios, we propose to consider reward instances encountered in applications such as dynamic pricing (e.g. those in [15] and [24]).
>
> * **On the paper clarity**. Thank you for pointing out that some sections may be too dense and for your suggestions to improve clarity. To help build intuition, we propose to add a worked example of the DP procedure on a simple line graph, such as the one in Figure 1, and a diagram to visually illustrate the procedure.
>
>
> [15] Kanishka Misra, Eric M. Schwartz, and Jacob Abernethy. Dynamic online pricing with incomplete information using multiarmed bandit experiments. Marketing Science, 38(2):226–333 252, 2019
>
> [24] Yining Wang, Boxiao Chen, and David Simchi-Levi. Multimodal dynamic pricing. Manage. Sci., 67(10):6136–6152, October 2021

---

### Decision · Program_Chairs · 2025-09-17

**Decision:**

Accept (poster)

**Comment:**

The paper presents a study on a stochastic multi-armed bandit variant in which the arms have an underlying tree graph structure with at most m modes (local maximums). The main contribution of the paper is to provide a polynomial-time algorithm to solve  the Graves-Lai optimization problem, which leads to an asymptotically optimal bandit algorithm.

The reviewers acknowledge the theoretical contribution of the paper, especially on the first computationally tractable solution for this setting. They also raise concerns, such as the graph has to be a tree graph and the experimental evaluation seems to be preliminary. The authors provided detailed responses, and the reviewers are in general satisfied with the authors' responses. In the end, all reviewers are positive toward the paper, but do not indicate very strong support. The paper is marginal to NeurIPS, but given the solid theoretical contribution, especially to encourage research on computationally tractable solutions for bandit algorithms, I believe the paper is worth to be published at NeurIPS. The authors should provide a thorough revision to address all reviewer concerns.

AC.